# Nanosecond pulsed electric fields induce cell-size-dependent selective permeabilization of urothelial cancer cells

Aleksander Kielbik [1] ✉, Emily Hellwich[2], Veronika Bahlinger[3], Pamela Sowa[4], Daniel Lambton[2], Markus Kühs[1], Maria Luisa Barcena[1], Olesya Vakhrusheva[1], Hendrik Proebsting [5], Simon Walz[1], Tilman E. Schäffer [2], Falko Fend [3], Vitalij Novickij[6,7], Bastian Amend [1] & Igor Tsaur[1]

Plasma membrane integrity is vital for cell viability, yet its controlled disruption enables targeted delivery of therapeutic agents. Here, we examined membrane durability and repair capacity in normal and malignant urothelial cells using short, high-voltage nanosecond pulses. Pulses were applied to monolayer cultures, spheroids, and patient-derived organoids. Plasma membrane permeability was assessed via YO-PRO-1 dye uptake, and mechanical effects of permeabilization were analyzed using atomic force microscopy. Urothelial cancer cells exhibited nearly fourfold higher dye uptake than non-malignant cells, along with more pronounced osmotic swelling and loss of cellular stiffness. Membrane resealing in cancer cells was delayed and exhibited stronger dependence on extracellular $Ca^{2+}$. The higher susceptibility of urothelial cells was correlated with their larger size, which enable them to reach the electroporation threshold at lower electric fields. These findings highlight key differences in membrane vulnerability and repair dynamics, providing foundation for the development of membrane-targeted therapies for urothelial cancer.

The plasma membrane maintains cellular homeostasis by regulating molecular transport between the cytoplasm and external environment. Preserving its integrity is fundamental for cell survival and proper physiological functions. Membranes continuously experience stress from physiological activities, such as muscle contractions, as well as from pathological factors, including lipid peroxidation, bacterial toxins, and traumatic injuries. In this context, the urothelium of the urinary bladder endures a challenging environment characterized by substantial stretch forces, osmotic and hydrostatic pressures, exposure to toxic substances, and susceptibility to microbial invasion. During bladder filling, distension triggers the mobilization of discoidal and fusiform-shaped vesicles through exocytosis, facilitating apical membrane expansion and thereby allowing controlled bladder accommodation without compromising structural integrity[1–4]. This process is facilitated by the cytoskeletal network of cytokeratins, which provides a crucial structural framework and elasticity, that allows cells to endure extreme stretching forces[2,5]. Additionally, uroplakins embedded in the apical membrane contribute to structural resilience and regulate the membrane permeability under mechanical stress[6].

In cancer cells, exocytosis is often dysregulated, thereby promoting tumor progression and metastasis[7–9]. Disruption of the cytoskeleton in urothelial cancer cells leads to cell softening and an increased metastatic potential[10–12]. Cancer progression has also been correlated with decreased levels of uroplakins in the urothelial cancer cells[13,14]. Furthermore, urothelial cancer cells show dysregulated expression of phospholipid-binding proteins from the annexin family, which play a central role in membrane repair[15–18]. Despite these known structural alterations, the differences in membrane durability and repair efficiency among urothelial cancer cells remain poorly understood.

Targeting the cell membrane of cancer cells has been used as a standalone therapy or a means of enhanced drug delivery with novel membrane disrupting agents[19–22], as well as in physical treatments such as sono- or electroporation[23,24]. Application of nanosecond pulsed electric fields (nsPEFs) increases the membrane permeability of living cells, forming nanometer-size membrane lesions. This transient permeabilization allows the passage of ions and small molecules, thereby affecting cellular function and viability[25,26]. Following electroporation cells initiate $Ca^{2+}$-independent

[1]Department of Urology, University Hospital Tuebingen, Tuebingen, Germany. [2]Institute of Applied Physics, University of Tuebingen, Tuebingen, Germany. [3]Institute of Pathology and Neuropathology and Comprehensive Cancer Center, University Hospital Tuebingen, Tuebingen, Germany. [4]Department of Cardiology and Angiology, University Hospital Tuebingen, Tuebingen, Germany. [5]University Hospital Tuebingen, Faculty of Medicine, Eberhard Karls University Tuebingen, Tuebingen, Germany. [6]Institute of High Magnetic Fields, Vilnius Gediminas Technical University, Vilnius, Lithuania. [7]Department of Immunology and Bioelectrochemistry, State Research Institute Centre for Innovative Medicine, Vilnius, Lithuania. ✉e-mail: Aleksander.kielbik@med.uni-tuebingen.de

and dependent membrane repair processes to restore integrity[27–29]. This recovery involves vesicular transport and membrane remodeling mechanisms including lysosomal exocytosis[30,31] and Annexin V-mediated repair[29].

Electroporation can be applied to generate nano-scale lesions without a chemical intervention enabling robust studies on membrane vulnerability and resealing capacity[32]. In this study we investigated the effect of nsPEF on plasma membrane integrity and the resealing dynamics of cancer and normal urothelial cells.

We assessed the membrane repair kinetics by time-lapse imaging of YO-PRO-1 (YP) uptake, a membrane-impermeant fluorescent dye, following exposure to nsPEFs across a broad range of electric field intensities, both in the presence and absence of extracellular Ca²⁺. Additionally, using patient-derived organoids (PDO) of urothelial cancer, we investigated the secondary effects of membrane permeabilization, such as cell swelling and loss of stiffness. Our findings demonstrate that urothelial cancer cells exhibit heightened susceptibility to plasma membrane disruption, as evidenced by more extensive damage and slower recovery following nsPEF exposure.

## Results

### Urothelial cancer cell membranes show increased sensitivity to electroporation compared to normal urothelial cells

The electroporation setup was established with a custom 3D-printed micromanipulator with tungsten electrodes positioned at a 45° angle to coverslip with the cell monolayer in physiological solution with YP dye and different concentration of Ca²⁺ (Fig. 1a). The electrode configuration produced the inhomogenous electric field enabling the observation of cell response in the wide spectrum of electric field intensities (Fig. 1b and supplementary Fig. S1a–e). Images of YP fluorescence were taken every 3 s for 180 s total. A series of 200, 300-ns pulse was applied at 10 s after starting the acquisition of images. The fluorescence time course (mean ± standard error of the mean (SEM)) was recorded in both normal urothelial cells and urothelial cancer cells.

A train of 200 pulses, each 300 ns long at 11.5 kV/cm, induced rapid YP uptake in all tested cell lines (Fig. 1c). Side-by-side experiments consistently demonstrated that T24 and UM-UC-3 cancer cells exhibited significantly greater membrane permeabilization than the normal urothelial cell lines SV-HUC-1 and HBLAK, as evidenced by a significantly larger area under the curve (AUC) of YP fluorescence across all tested extracellular Ca²⁺ concentrations ($p < 0.0001$ for all comparisons between cancer and normal urothelial cell lines). Cells not treated with nsPEFs showed no detectable increase in YP fluorescence over the observation period (as shown in supplementary Fig. S2a).

On average, 3 min after electroporation with 200 pulses of 300 ns at 11.5 kV/cm, YP fluorescence was 2- to 6-fold higher in cancer urothelial cells compared to normal urothelial cells under all tested conditions ($p < 0.05$) (Fig. 1d). In the 0 mM Ca²⁺, YP fluorescence was 3–5.1 times higher in T24 and UM-UC-3 urothelial cancer cells compared to both normal HBLAK and SV-HUC-1 cells ($p < 0.001$ for all comparisons).

Despite the reduction in overall YP uptake caused by extracellular Ca²⁺, the significant difference in membrane permeabilization between cancer and normal cells was preserved. In the presence of 2 mM Ca²⁺,

T24 cells exhibited YP fluorescence 2.4-fold higher than HBLAK cells ($p < 0.05$) and 4.8-fold higher than SV-HUC-1 cells ($p < 0.001$). Similarly, at 5 mM Ca²⁺, YP uptake measured 3 min post-exposure in T24 cells was 4.8-fold greater than in HBLAK cells ($p < 0.0001$) and 6.1-fold greater than in SV-HUC-1 cells ($p < 0.0001$). In cancer UM-UC-3 cells, YP uptake in 2 mM Ca²⁺ was 4.3 times higher than in normal SV-HUC-1 cells ($p < 0.0001$) and 2.1 times higher than in HBLAK cells ($p < 0.001$). Likewise, in the presence of 5 mM Ca²⁺, UM-UC-3 cells showed 4.5-fold higher fluorescence than SV-HUC-1 cells ($p < 0.01$) and 3.6-fold higher than HBLAK cells ($p < 0.01$).

The pronounced differences in membrane permeabilization between cancer and normal urothelial cells may arise from a heightened sensitivity of cancer cell membranes to electric field-induced disruption, resulting in larger or more sustained membrane defects. Alternatively, they may reflect a diminished capacity for membrane repair in cancer cells. To investigate these potential mechanisms, we performed a kinetic analysis of YP uptake.

### Urothelial cancer cells exhibit a heightened dependence on extracellular Ca²⁺ for efficient membrane repair compared to normal cells

Recent reports indicate that electroporated cells are capable of repairing membrane defects regardless of the applied voltage, and that electroporation is unlikely to directly cause irreversible membrane rupture[27]. Cellular membrane repair involves both Ca²⁺-dependent and -independent mechanisms[28,29]. To investigate the role of extracellular Ca²⁺ in facilitating membrane repair, we electroporated cells in physiological solutions containing 0 mM, 2 mM, or 5 mM Ca²⁺.

The average YP uptake over time was fitted with a single-exponential function, which was extrapolated beyond the experimental observation period ($R^2 > 0.95$ for all cell lines in all tested conditions) (Fig. 2a). The results demonstrated that YP fluorescence intensity was directly dependent on the electric field strength, with higher voltages causing greater cell membrane disruption (Fig. 2b). In the absence of external Ca²⁺, as well as in 2 mM and 5 mM Ca²⁺ solutions, decrease of electric field from 11.5 kV/cm to 10 kV/cm led to a 1.2–2-fold reduction in dye uptake three minutes post-exposure (Fig. 2b). A further reduction to 8 kV/cm resulted in a mean 3-fold decrease in uptake across all tested cell lines (For more details see additional data provided in supplementary Fig. S2b).

The exponential function of YP fluorescence is characterized by the time constant $\tau$, which quantifies the rate at which membrane recovery progresses following electroporation (Fig. 2a). In the T24 cell line at 11.5 kV/cm, $\tau$ decreased significantly from 130 s to 84 s in the presence of 2 mM Ca²⁺ ($p < 0.001$) and to 86 s in 5 mM Ca²⁺ ($p < 0.001$) (Fig. 2c). Similarly, in the UM-UC-3 cell line, $\tau$ was reduced from 115 s to 75 s in both 2 mM and 5 mM Ca²⁺ ($p < 0.001$). Despite this clear trend, the decrease in $\tau$ in the SV-HUC-1 and HBLAK normal urothelial cell lines with increasing Ca²⁺ concentrations was not statistically significant. A similar pattern in all tested cell lines was observed at 10 kV/cm and 8 kV/cm, further reinforcing that urothelial cancer cells exhibit a stronger dependence on Ca²⁺-mediated membrane repair mechanisms. The effect of extracellular Ca²⁺, more pronounced in urothelial cancer cells, was further evidenced by a reduction in YP fluorescence changes measured at 3-s intervals following exposure, across all tested cell lines and independent of electric field strength (see supplementary Fig. S3a for details)

Interestingly, higher electric fields did not significantly alter the $\tau$ of the exponential fits indicating that membrane repair mechanisms were proportionally activated based on the extent of damage, allowing the membrane to be restored within a similar timeframe (supplementary Fig. S3b). This shows that increased electric field intensity of nsPEFs causes greater cell membrane damage but does not affect their repair capacity.

Although urothelial cancer cell lines generally exhibited less efficient and more Ca²⁺-dependent membrane repair mechanisms, this alone is unlikely to fully account for the differences in YP uptake between normal and malignant urothelial cells observed after nsPEF exposure across varying Ca²⁺ concentrations. Therefore, further investigations were conducted to identify additional factors contributing to the selective permeabilization observed in the initial experiments.

### The larger size of urothelial cancer cells, compared to normal cells, increases their susceptibility to permeabilization at lower electric field intensities

The transmembrane potential (TMP) evoked by the electric field can be influenced by the cells shape and size. Spherical cells generally exhibit a uniform TMP distribution, while irregularly shaped cells display localized regions of enhanced TMP, which can affect electroporation efficiency and uptake of molecules[33]. Theoretical studies particularly those relying on the electric circuit modeling, suggested that larger cells develop larger TMP when placed in homogenous electric filed[34]. The experimental data confirms that the more severe

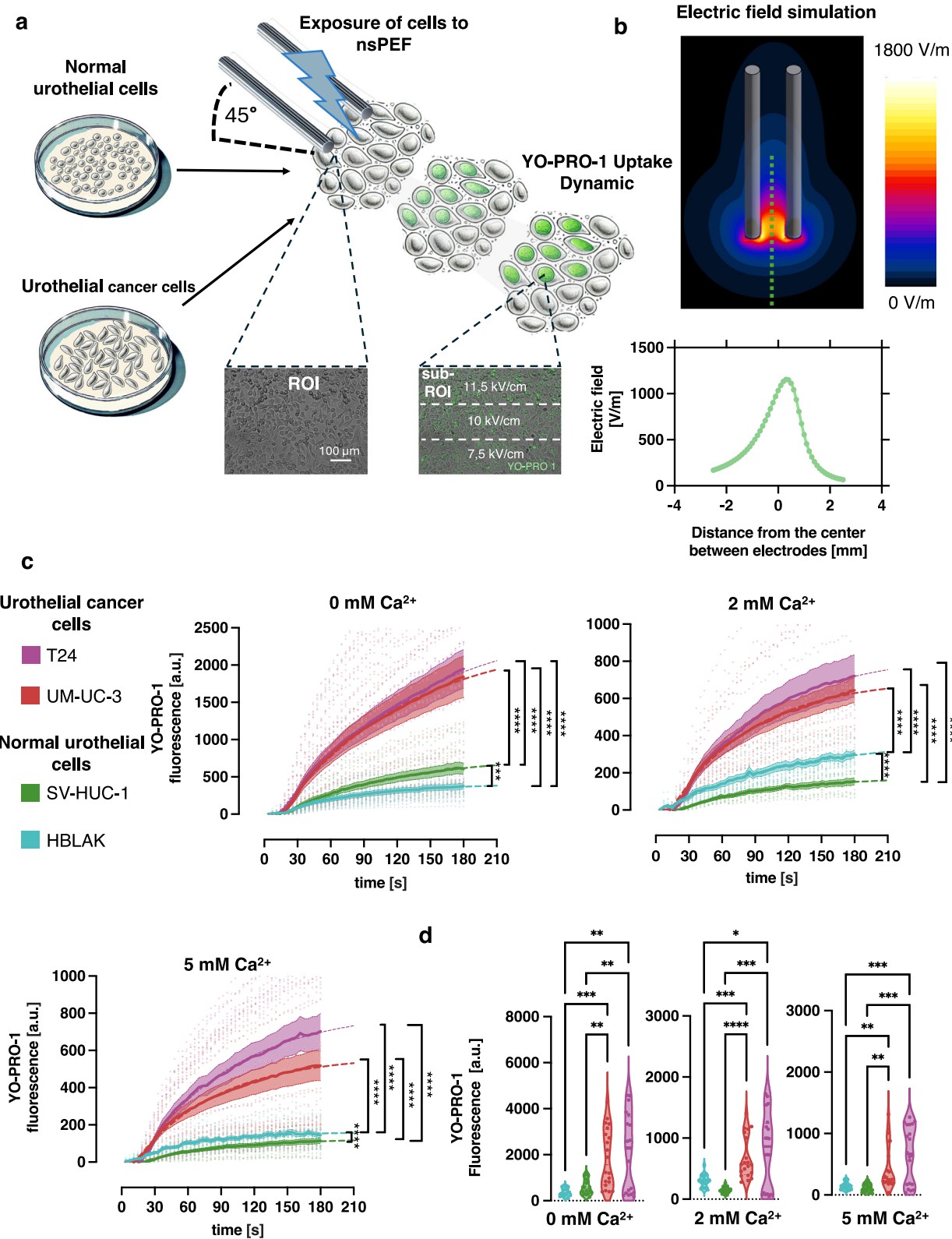

electroporation damage is done to the larger cells[34–36]. To assess whether morphological differences contribute to the increased permeabilization observed in urothelial cancer cells, we analyzed previously studied cells for key shape and size parameters, including cell area, major and minor axis lengths, and form factor, following cytoskeletal staining with phalloidin (Fig. 3a).

Our results revealed a significant difference in cell size between urothelial cancer cells (T24 and UM-UC-3) and normal urothelial cells (SV-HUC-1 and HBLAK) (Fig. 3b). On average, urothelial cancer cells were 1.8 times larger in area compared to normal urothelial cells ($p < 0.0001$ for each normal vs. cancer cell line comparison). Similarly, the perimeter was 1.4 times longer, while the major and minor axes were 1.3 and 1.4 times longer,

**Fig. 1 | Evaluation of cell membrane electropermeabilization by YO-PRO-1 (YP) dye uptake. a** A schematic of the setup that enables a fluorescence imaging of electroporated cell. Tungsten electrodes were positioned on the glass coverslip with seeded normal or cancer urothelial cells on the stable 45° angle between coverslip and electrodes. An 800 × 600 μm region of interest (ROI) was positioned such that its longer edge overlapped the line connecting the tips of the two electrodes. The ROI extended beyond the electrode gap on the side opposite the electrodes. ROI was divided into three equal subregions (subROI), each measuring 800 × 200 μm, by drawing two parallel lines, evenly spaced along the shorter axis of the ROI. **b** Calculated electric field distribution along the line perpendicular to the axis connecting the centers of both electrodes with 1 V applied between them. The ROI were placed in the electric field marked in (**c**). The time course of YP fluorescence measure as arbitrary units (a.u.) (mean ± the standard error of the mean (SEM)) in T24, UM-UC-3, SV-HUC-1 and HBLAK cells after electroporation by train of 200, 300-ns, 11.5 kV/cm at 10 Hz pulse in the solution with 1 μM YP and at different Ca$^{2+}$

concentration (marked in the titles) $n = 18$–20. The mean fluorescence was fitted with single-exponential curve (coefficients of determination (R$^2$) > 0.95 for all tested conditions and cell lines). The comparison of area under curve (AUC) of YP fluoresce curves show highly significant differences between normal and cancer urothelial cells. Differences were assessed using one-way Welch's ANOVA W (DFn; DFd). For 0 mM Ca$^{2+}$: W(3, 36.07) = 56.15; for 2 mM Ca$^{2+}$: W(3, 36.13) = 93.29; and for 5 mM Ca$^{2+}$: W(3, 37.59) = 72.11. Dunnett's T3 post hoc test was used to correct for multiple comparisons. Statistical significance is indicated as follows: ($p \geq 0.05$), *$p < 0.05$, **$p < 0.01$, ***$p < 0.001$, ****$p < 0.0001$. **d** Violin plot showing the distribution of mean YP fluorescence intensity at 180 seconds (s) after the exposure in 0 mM, 2 mM and 5 mM concentration of Ca$^{2+}$ $n = 18$–20. Differences were assessed using one-way Welch's ANOVA followed by Dunnett's T3 post hoc test for multiple comparisons: for 0 mM Ca$^{2+}$: W(3, 35.06) = 16.22; for 2 mM Ca$^{2+}$: W(3, 35.62) = 31.60; and for 5 mM Ca$^{2+}$: W(3, 37.66) = 13.98.

respectively ($p < 0.0001$ for each comparison). However, no statistically significant difference was observed in the form factor, an index of cell circularity, suggesting that while cancer cells are larger, their overall shape remains comparable to that of normal urothelial cells.

As a next step in evaluating whether the observed differences in cell size may influence electroporation efficiency, we calculated the electric field required to reach the transmembrane potential threshold for permeabilization as a function of pulse duration across cells with diameters ranging from 10 to 40 μm (Fig. 3c).

The relationship between cell size and electroporation threshold can be explained using basic capacitor principles, by modeling the cell membrane as a simple dielectric layer in a spherical capacitor. The transmembrane potential required to induce electroporation has been experimentally established at approximately 0.2 V for various cell types[27,37,38]. The external electric field E (in V/m) required to induce a transmembrane voltage $\Delta V$ across the cell membrane is described by the steady-state Schwan equation[39]:

$$E = \frac{2}{3} \times \left(\frac{\Delta V}{R}\right)$$

Where E is the external electric field, $\Delta V$ the induced transmembrane potential and $R$ the cell radius, t the time since the start of the pulse (in s) and $\tau$ the membrane charging time constant.

Because membrane charging is a time-dependent process, the induced transmembrane potential increases gradually during the pulse. This dynamic can be described by an exponential function:

$$\Delta V(t) = \Delta V_{\max} \times \left(1 - \exp\left(-\frac{t}{\tau}\right)\right)$$

where t is the time since the start of the pulse and τ is the membrane charging time constant (2 μs for mammalian cells). Using this model, we calculated the electric field strength required to reach a transmembrane potential of 0.2 V across various cell sizes and pulse durations. The results demonstrate that larger cells require lower electric field intensities to reach the electroporation threshold. In the context of urothelial cells, the electric field needed to permeabilize a normal urothelial cell with an average diameter of 20 μm is approximately 1.5 times higher than that required for a cancer cell with an average diameter of 30 μm. Moreover, the thresholds for cells of different diameters remain consistently spaced across the full range of pulse durations studied. This suggests that the relative influence of cell size on the electroporation threshold remains stable, regardless of the pulse duration.

The difference in cell size between non-malignant and cancer urothelial cells was confirmed in a large-scale analysis using nine distinct tissue microarrays from patients with urothelial cancer (Fig. 3d). The dataset included 47 samples of normal urothelial tissue, 66 samples of cancer cells from primary bladder tumors, and 56 samples from lymph node metastases. After HE staining, a skilled pathologist evaluated the samples and distinguished cancer and normal urothelial cells. Once the areas of normal and

cancer urothelial cells within the tissue section were marked, we measured the 2D cross-section area and form factor ("roundness") of single cells, comparing normal and cancer urothelial cells (see supplementary Fig. S4a–d for details). The analysis confirmed that the mean single cell area in tissue sections was 111.8 ± 2.5 μm² for urothelial cancer cells from both primary tumors and lymph node metastases, compared to 90.7 ± 3.1 μm² for normal urothelial cells, indicating that cancer cells were approximately 23% larger ($\Delta = 21.1$ μm²) ($p < 0.001$ for both comparisons). Moreover, a small but statistically significant difference in cell form factor was observed. Urothelial cancer cells exhibited a mean form factor of 0.815 ± 0.0009, compared to 0.804 ± 0.0011 in normal urothelial cells ($p < 0.05$ for both comparisons).

Based on our calculations, the characteristically larger size of urothelial cancer cells helps to explain the enhanced electroporation efficiency observed in our monolayer experiments. This difference in cell size not only applied to the specific cell lines used in our in vitro model but was also confirmed in histopathological samples from urothelial cancer patients.

With the evidence of enhanced permeabilization in larger cancer cells under monolayer conditions, we transitioned to a three-dimensional model to evaluate the consistency of this effect. In 3D cell structures such as spheroids, the effective membrane charging time constant $\tau$ may increase due to local extracellular resistance and intercellular coupling, potentially influencing electroporation dynamics. Therefore, in the next stage of our study, we established a 3D model using patient-derived urothelial cancer cells, designed to preserve both their physiological characteristics and a controlled homogeneous, spheroidal architecture. This system enabled a direct comparison of membrane permeability across different samples following exposure to nsPEFs.

### Validation of PDOs and spheroids from the cell culture

PDOs are increasingly used for large-scale, personalized testing of drug responses and treatment modalities. These advanced models provide insights into patient-specific resistance and susceptibility to therapies, offering the potential to guide individualized treatment decisions[40]. To investigate membrane permeabilization, PDOs were used to capture tumor-specific variability in susceptibility to nsPEFs.

Primary bladder cancer cells were isolated directly from the bladder epithelium of patients undergoing transurethral resection of tumor or radical cystectomy for muscle-invasive bladder cancer (see supplementary Table S1 for more details). The resulting human PDOs were cultured and successfully passaged more than eight times. To allow for comparative analysis, we also generated 3D spheroids from non-muscle-invasive urothelial cancer cells (RT4) and from the immortalized normal urothelial cell line SV-HUC-1. From the primary PDOs cultures, we selected those samples that reliably formed structurally robust spheroids, defined by an average form factor greater than 0.6 and average diameter longer than 100 μm in 3-week culture (Fig. 4a). These criteria ensured the production of uniform and physiologically relevant spheroids, suitable for electroporation testing using our customized experimental setup previously applied in monolayer studies.

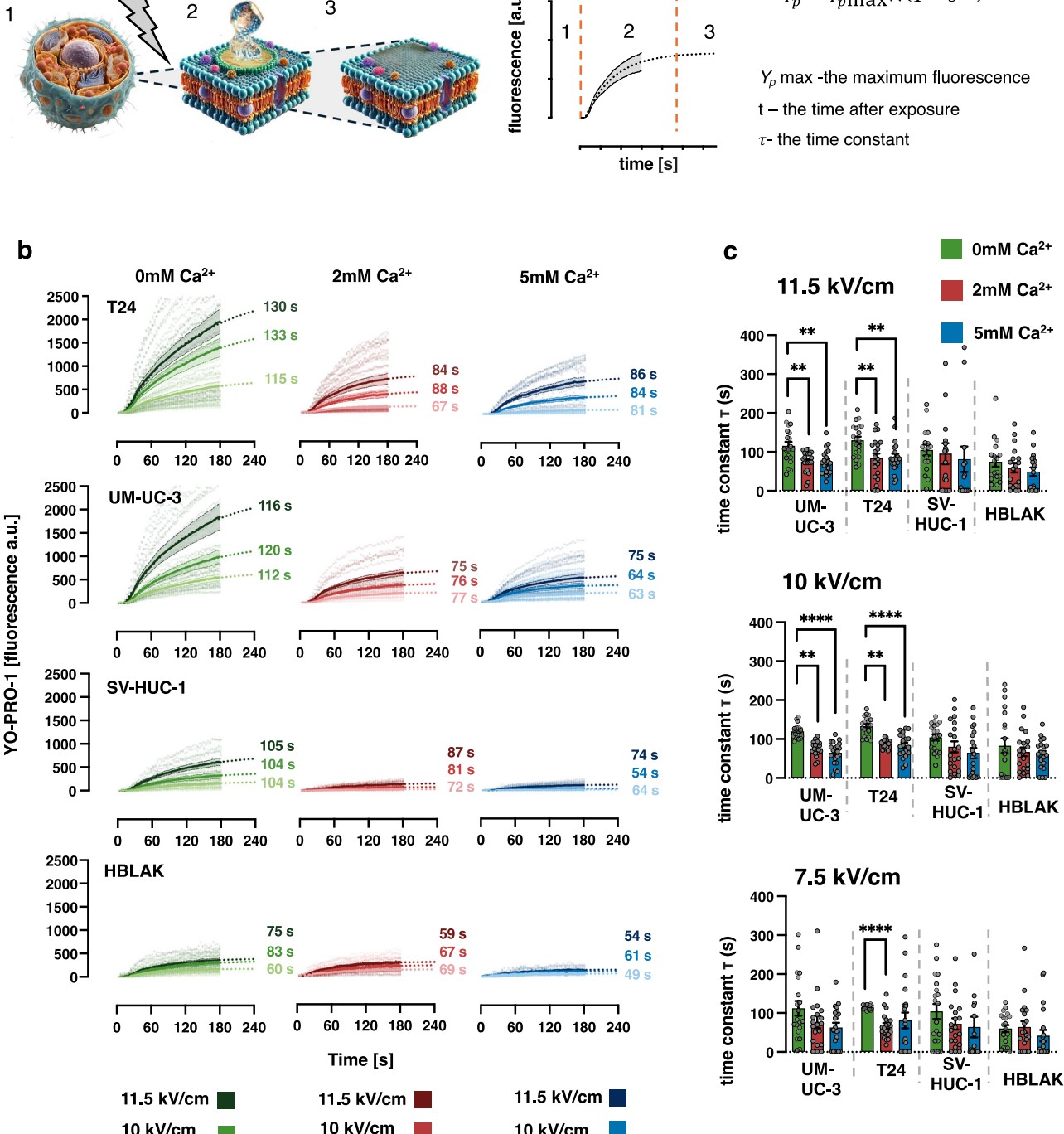

As a result, PDOs from three donors were selected for further analysis. Morphological profiling of the spheroids used in subsequent electroporation experiments revealed no significant differences in size or structure, except for PDO 319, which showed a statistically significant smaller size when compared to spheroids derived from non-malignant urothelial SV-HUC-1 cells ($2.44 \times 10^5 \pm 0.29 \times 10^5\ \mu m^2$ vs. $1.30 \times 10^5 \pm 0.14 \times 10^5\ \mu m^2$; $p < 0.01$) (Fig. 4a).

The selected spheroids were subjected to histopathological evaluation to confirm that PDOs were composed of urothelial cancer cells (Fig. 4b). Immunohistochemical analysis further characterized the tumor subtypes: PDO 270 exhibited a luminal phenotype, PDO

319 a basal subtype, and PDO 154 was identified as a double-negative neuroendocrine tumor. Importantly, the expression profile of key urothelial cancer markers in the PDOs remained consistent with those observed in the corresponding primary tumor tissues (see supplementary Fig. S5a, b for more details).

### PDOs show higher membrane permeabilization after exposure to nsPEFs compared to spheroids from normal urothelial cells

Electroporation was performed on PDOs and spheroids using nanosecond pulses delivered between two parallel tungsten rod electrodes. The electrodes were positioned such that the front of the spheroid was precisely

**Fig. 2 | Urothelial cancer cells exhibit a heightened dependence on extracellular $Ca^{2+}$ for efficient membrane repair compared to normal cells. a** Schematic representation of membrane resealing dynamics, visualized through changes in YP fluorescence. (1). Prior to electroporation, the cell membrane remains intact, preventing the uptake of the membrane-impermeant dye YP. (2) Following electroporation, transient nanopores form in the plasma membrane, enabling the entry of YP and water from the extracellular environment into the cell. The kinetics of dye uptake can be fitted with a single-phase exponential curve, reflecting the dynamics of membrane permeability. (3) As the membrane reseals, dye uptake ceases and the fluorescence signal reaches a plateau. This plateau indicates successful membrane recovery, with no further entry of YP. **b** Time course of YP dye uptake in urothelial cancer cells and normal urothelial cells under three different extracellular $Ca^{2+}$ concentrations: 0, 2, and 5 mM. The uptake dynamics were fitted with a single-phase exponential model, and the time constants τ (in s) are indicated next to each curve.

The fits showed a strong correlation, with $R^2$ exceeding 0.95 for all cell lines under all tested conditions, indicating robust exponential uptake kinetics. Data are presented as mean ± SEM, with $n = 18–20$ per condition. **c** Average time constants derived from the exponential fits, shown as mean ± SEM ($n = 18–20$). Points represent individual measurements from independent experimental replicates. Differences between different $Ca^{2+}$ concentration for each cell line and electric field intensity were assessed using one-way Welch's ANOVA for followed by Dunnett's T3 post hoc test for multiple comparisons. Welch's ANOVA results (W (DFn, DFd)) for comparisons across $Ca^{2+}$ concentrations at 11.5 kV/cm, 10 kV/cm, and 7.5 kV/cm were as follows: For HBLAK: W(2.000, 35.04) = 1.125; W(2.000, 32.86) = 0.5627; W(2.000, 33.46) = 0.6787. For SV-HUC-1: W(2.000, 32.85) = 0.5254; W(2.000, 32.32) = 1.099; W(2.000, 33.62) = 1.085. For T24: W(2.000, 37.64) = 6.939; W(2.000, 36.92) = 14.82; W(2.000, 28.14) = 22.13. For UM-UC-3: W(2.000, 36.72) = 6.800; W(2.000, 36.92) = 14.82; W(2.000, 37.80) = 2.404.

aligned with the midpoint of the line connecting the two electrodes, ensuring its placement within the region of highest electric field intensity (Fig. 5a and supplementary Fig. S6). Within the region occupied by the spheroid, the electric field strength varied by no more than 10% between its maximum and minimum values. YP uptake was monitored for 10 min post-exposure (Fig. 5b). During the 10-min observation period, no significant increase in YP fluorescence was observed in spheroids incubated in 0, 2, or 5 mM medium when electroporation was not performed (see supplementary Fig. S7a). For statistical comparison of YP uptake across conditions, we calculated AUC of fluorescence intensity–time curve.

At 10 min post-exposure in 0 mM $Ca^{2+}$, the final YP fluorescence level in in all tested PDOs was 2–3 times higher than that measured in spheroids derived from normal urothelial SV-HUC-1 cells ($p < 0.0001$ for all comparisons with SV-HUC-1 spheroids) (Fig. 5c. supplementary Fig. S7b, c). A similar trend was observed in the presence of 2 mM $Ca^{2+}$ with final fluorescence 1.5 to 3.5 times higher in PDO compared to spheroids from normal urothelial cells ($p < 0.05$). Under 5 mM $Ca^{2+}$ condition, PDO 270 showed a modest 1.2-fold higher YP uptake, which was not significantly different from normal cell spheroids ($p > 0.05$), whereas other PDOs still exhibited 1.8 to 3.9 times higher YP fluorescence than SV-HUC-1 cells ($p < 0.0001$).

All PDOs and spheroids derived from RT4 and SV-HUC-1 cells exhibited a $Ca^{2+}$-dependent decrease in YP fluorescence (supplementary Fig. S7d). On average, fluorescence measured 10 min after exposure decreased by 1.6-fold in the presence of 2 mM $Ca^{2+}$ and by 1.8-fold with 5 mM $Ca^{2+}$. Notably, increasing $Ca^{2+}$ from 2 mM to 5 mM reduced fluorescence in all spheroids except RT4, which showed a 1.2-fold increase.

YP uptake findings align with observations from cell monolayer permeabilization studies, highlighting that nsPEFs induce significantly greater membrane disruption in urothelial cancer cells than in non-malignant counterparts. Linear regression analysis revealed a statistically significant positive correlation between mean cell size and YP uptake after exposure in 2 mM $Ca^{2+}$ solution ($r = 0.36$, $r^2 = 0.13$, $p = 0.0148$) (Fig. 5d). Despite considerable variability in fluorescence intensity among spheroids of the same type, spheroids composed of larger cells consistently exhibited higher levels of permeabilization. No significant correlation was observed at 0 or 5 mM $Ca^{2+}$, suggesting that extremes of extracellular calcium attenuate the size-dependent effect on membrane permeabilization, through cell type–specific interactions between $Ca^{2+}$ and membrane repair (supplementary Fig. S7e).

The differences in YP uptake observed between various spheroids and PDOs could not be attributed to variations in whole spheroid size. Linear regression analysis revealed no significant correlation between spheroid diameter and YP fluorescence intensity measured 10 min after exposure at any of the tested $Ca^{2+}$ concentrations (supplementary Fig. S7e). These results show that, in three-dimensional spheroids, membrane permeabilization is governed primarily by the size of individual cells rather than by the overall spheroid dimensions.

## Disruption of plasma membrane causes more robust cell swelling and the loss of cell stiffness of urothelial cancer cells

Following membrane permeabilization, cells undergo colloid osmotic swelling as a result of the osmotic gradient established across the compromised cell membrane[41,42]. To investigate whether the superior YP entry into cancer cells correlates with increased swelling, we quantified the progression of 2D projected area of spheroid after exposure to nsPEFs (Fig. 6a). We employed machine learning–based image analysis to detect the boundaries of the spheroid in each frame of the time-lapse acquisition and to track changes in its size over the 10-minute period following exposure to nsPEFs (see supplementary Fig. S7 for details). To statistically compare the progression of 2D projected area over time, we calculated the AUC for each size–time curve and compared the AUCs of RT4 spheroids and PDOs to those of non-malignant SV-HUC-1 spheroids (for the comparison of the percentage increase in spheroid size 10 min after exposure, see supplementary Fig S8a). In line with the YP uptake results, PDOs showed greater increases in projected 2D area compared to spheroids derived from normal urothelial cells.

Specifically, the 2D projected area of PDOs in 0 mM $Ca^{2+}$ increased from 14 to 21% whereas in RT4 spheroids the area expanded by 42%. In contrast, spheroids from normal urothelial SV-HUC-1 cells showed only 12% increase in 2D projected area— significantly lower than all PDOs and RT4 spheroids ($p < 0.001$ for all comparisons with SV-HUC-1 spheroids). Also in the presence of 2 mM and 5 mM $Ca^{2+}$, all urothelial cancer spheroid types exhibited a significantly greater size increase compared to non-malignant SV-HUC-1 spheroids ($p < 0.0001$ for all comparisons).

Interestingly, reduced membrane permeabilization in $Ca^{2+}$-containing medium did not necessarily prevent spheroid swelling (supplementary Fig. S8b). RT4 and SV-HUC 1 spheroids showed a significantly lower increase in 2D projected area at 2 and 5 mM $Ca^{2+}$ than under 0 mM $Ca^{2+}$ conditions ($p < 0.0001$ and $p < 0.001$ for RT4; $p < 0.001$ and $p < 0.05$ for SV-HUC-1, comparing AUC in 5 mM vs AUC in 0 mM and 2 mM $Ca^{2+}$, respectively). In PDO 154, no statistically significant difference in spheroid size progression was observed across $Ca^{2+}$ concentrations and the increase in 2D projected area of PDO 270 and PDO 319 was significantly greater in 5 mM $Ca^{2+}$ compared to 0 mM and 2 mM $Ca^{2+}$ ($p < 0.001$ for both comparisons). These distinct effects of extracellular $Ca^{2+}$ may reflect its variable impact on the mechanical properties of cells following intracellular accumulation. To further explore the mechanical changes occurring in spheroids after electroporation, we developed a custom setup enabling rapid AFM force mapping after nsPEF exposure (Fig. 6b).

AFM measurements commenced 60 s post-exposure, with data acquisition performed every 3 min up to 10 min. For compatibility with AFM, electrodes were carefully removed using a micromanipulator within s after nsPEF exposure. Mechanical changes were assessed by mapping stiffness at 25 points (5 × 5 grid) across the spheroid surface, and corresponding topographical images were recorded over time (Fig. 6b). To obtain reproducible and reliable AFM measurements, stable adhesion of the spheroid to the substrate was essential. This was successfully achieved with

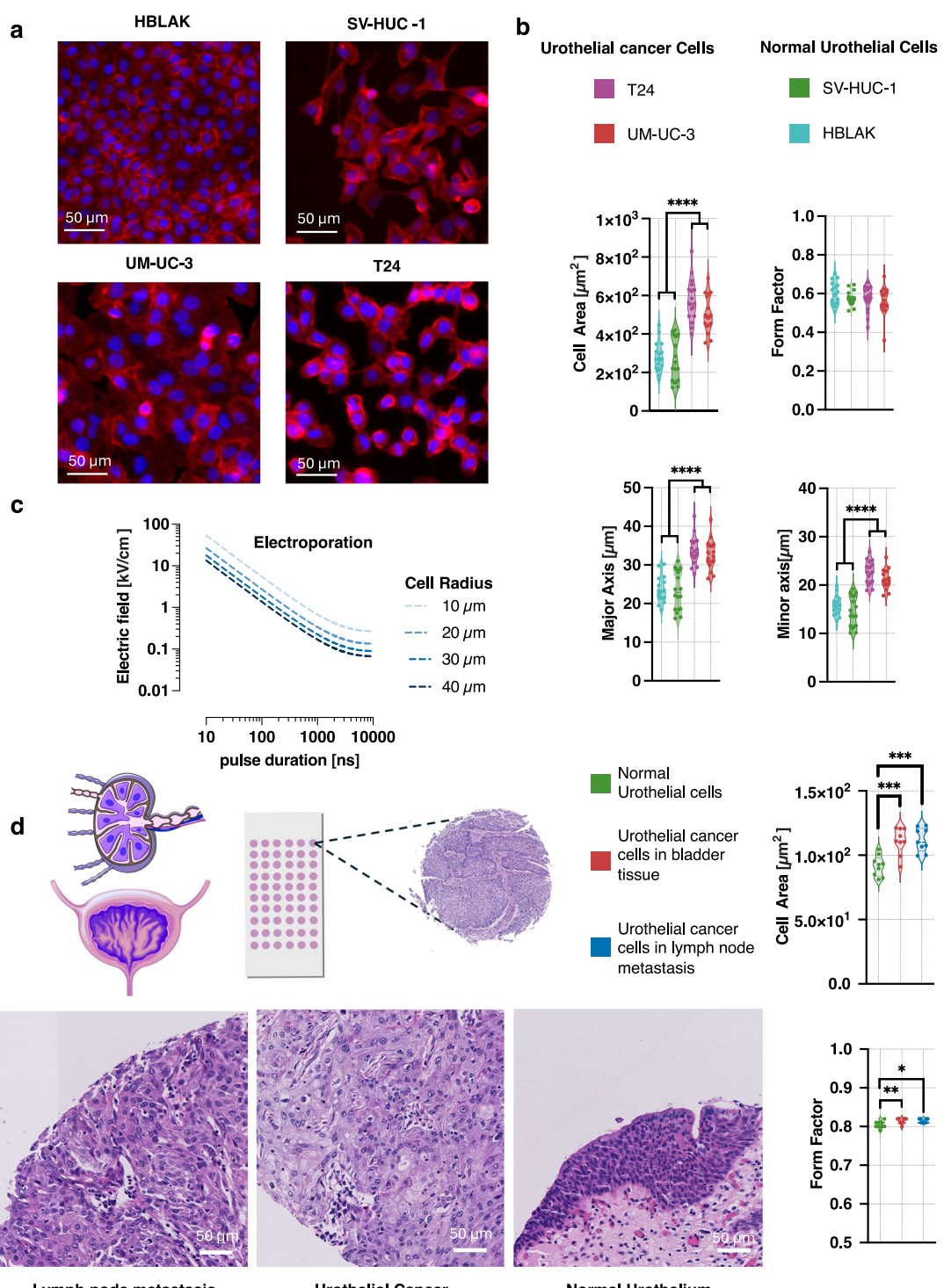

spheroids derived from both RT4 cancer cells and normal urothelial SV-HUC-1 cells, which showed no signs of detachment or loss of stiffness without exposure to nsPEFs during repeated measurements over a 10-min period.

Prior to nsPEF exposure, the Young's modulus ("stiffness") in live spheroids derived from RT4 and SV-HUC-1 cells showed no significant difference between the two groups under either 0 mM or 2 mM $Ca^{2+}$ conditions (Fig. 6c). However, 10 min after exposure in 0 mM $Ca^{2+}$, RT4 exhibited a significant reduction in stiffness to 401.8 ± 33 Pa (mean ± SEM), approximately three times lower than SV-HUC-1 spheroids (1218.5 ± 211.2 Pa) $p < 0.0001$. A similar difference was observed in 2 mM

$Ca^{2+}$, where RT4 spheroids remained 2.7 times less stiff than SV-HUC-1 spheroids ($p < 0.001$).

The reduction in stiffness followed an exponential decay, where the most rapid decline occurred within the first 4 min post-exposure and stabilized near an equilibrium at 10 min (Fig. 6d). In 0 mM $Ca^{2+}$, statistically significant differences in normalized stiffness between normal and cancer spheroids were already apparent 4 min post-exposure: RT4 spheroids lost 50% ± 4.5% of their initial stiffness, while SV-HUC-1 spheroids showed a 26% ± 6% reduction ($p < 0.05$). By 10 min, the stiffness of SV-HUC-1 spheroids had decreased by 29%, whereas RT4 spheroids demonstrated a 59% reduction ($p < 0.05$). Interestingly, the presence of $Ca^{2+}$ further reduced

**Fig. 3 | Impact of urothelial cell size on response to nsPEF exposure.**
**a** Representative images of monolayers of HBLAK, SV-HUC-1, UM-UC-3, and T24 cells stained with Phalloidin and Hoechst 33342 dyes to visualize actin filaments (red) and nuclei (blue), respectively. **b** Morphological features of the cells area measurements were obtained from 18 randomly selected ROIs (each 250 μm × 250 μm) containing approximately 40–80 stained cells. The average of measured morphological feature of cells within each ROI was treated as an independent data point for statistical analysis, resulting in $n = 18$. Data were visualized using violin plots, and statistical comparisons between cell lines were performed using Welch's ANOVA, followed by Dunnett's T3 post hoc testing. Welch's ANOVA revealed statistically significant differences between cell lines in average cell area ($W(3, 37.28) = 46.01$), major axis length ($W(3, 37.46) = 38.88$), and minor axis length ($W(3, 37.32) = 49.28$). No significant differences were observed in the form factor ($W(3, 36.34) = 0.98$). **c** Plot of the electroporation threshold electric field as a function of pulse duration for cells of varying radii. The external electric field (E, in V/m) necessary to reach electroporation threshold was calculated for the cathode-facing pole of a spherical cell, assuming a membrane charging time constant (τ) of 2 μs. See main text for methodological details. **d** Representative images and violin plots showing the average cell area and form factor of single cells across 9 tissue microarrays (TMAs), with each point representing the mean value per TMA ($n = 9$). The dataset includes 47 normal urothelial tissue samples, 66 samples from primary urothelial carcinomas, and 56 samples from lymph node metastases. Statistical comparison of mean urothelial cell area across normal urothelial tissue, urothelial carcinoma, and lymph node metastasis was performed using weighted one-way ANOVA, with the number of tissue sections per array used as weights and the Tukey adjustment for multiple comparisons. Pairwise comparisons using weighted marginal means revealed significantly larger mean cell areas in normal urothelial tissue compared to both cancer and lymph node tissue ($F(2, 24) = 13.42$) as well as lower form factor of normal urothelial cells, ($F(2, 24) = 6.39$).

the disparity in mechanical properties between cancer and normal spheroids. In 2 mM $Ca^{2+}$, SV-HUC-1 spheroids exhibited a 52% reduction in stiffness, while RT4 spheroids showed a 64% decrease within 10 min post-exposure. As a result, the difference in stiffness between the two groups was no longer statistically significant ($p > 0.05$).

## Discussion

Our research revealed a pronounced difference in plasma membrane sensitivity to nsPEFs between cancer and normal human urothelial cells. We attribute this disparity primarily to the larger size of urothelial cancer cells, which leads to a higher induced TMP, thereby facilitating electroporation at lower electric field intensities. The specific targeting of urothelial cancer cells based on their larger size represents a promising therapeutic strategy that is not utilized by any currently available treatments. This approach holds potential for effective permeabilization of tumor while minimizing off-target membrane permeabilization of non-malignant urothelial tissue. The lack of tumor-specific targeting in current intravesical therapies highlights the limitations of current treatment approaches for urothelial cancer[43,44]. While often effective, intravesical BCG instillation induces a broad local immune response, which can lead to adverse effects such as immune cystitis and result in therapy interruption or discontinuation in approximately 30% of patients[44]. After exposure to nsPEFs during the permeabilized state urothelial cancer cells are likely to be more susceptible to chemotherapeutic agents or gene transfer, offering a therapeutic opportunity to selectively target malignant cells while sparing non-malignant urothelial tissue.

While our research demonstrated that differences in cell size play a significant role in the selective targeting of urothelial cancer cells, previous studies have also shown that membrane cholesterol content[45,46] and the expression of specific membrane proteins[47] can influence sensitivity to membrane permeabilization. Therefore, variations in the lipid and protein composition of urothelial cancer cells may also contribute to the observed differences in electroporation efficacy.

In this study, we compared membrane repair efficacy between normal and cancer urothelial cells with a focus on the role of extracellular $Ca^{2+}$ concentration. Analysis of YP uptake kinetics revealed that membrane resealing in cancer cells is more dependent on extracellular $Ca^{2+}$. Membrane repair following electroporation can occur via both $Ca^{2+}$-dependent and $Ca^{2+}$-independent mechanisms[28]. The shift toward $Ca^{2+}$-dependent repair observed in cancer cells may represent a potential therapeutic target, extending the differential susceptibility and duration of permeability between cancer and normal urothelial cells.

Although electroporation is a transient process, it may ultimately lead to cell death due to $Ca^{2+}$ overload, mitochondrial dysfunction, ROS production, and ATP depletion[48–51]. Therefore, while extracellular $Ca^{2+}$ enhance membrane resealing, it may also promote delayed cell death[52]. The overall cytotoxic effect may thus depend on cell's capacity to restore intracellular $Ca^{2+}$ levels or osmotic balance, and not exactly mirror the initial extent of membrane damage alone[53].

Our study revealed not only superior permeabilization of urothelial cancer cells, but also more pronounced secondary effects such as osmotic swelling and stiffness reduction. Osmotic swelling occurs as a result of Donnan-type colloid osmotic pressure[54]. Research has confirmed that this process can be blocked with solutes too large to pass the pore, indicating that it is directly linked to cell membrane permeabilization[55]. Our data show that although extracellular $Ca^{2+}$ promotes membrane resealing, it also exerts a variable effect on cell swelling following permeabilization. Notably, previous studies have also reported that in solutions containing pore-blocking large solutes and extracellular $Ca^{2+}$, microtubule depolymerization occurs even in the absence of cell swelling or blebbing[31]. The effect of $Ca^{2+}$ on osmotic swelling varied significantly among the tested spheroids and PDOs, underscoring its inconsistent and less predictable influence on cellular downstream effects following electroporation.

While cell swelling may have influenced the average fluorescence intensity within spheroids, this effect is expected to be minor. The effect of extracellular $Ca^{2+}$ on cell swelling varied among spheroid and PDO types, whereas YP fluorescence intensity generally decreased as $Ca^{2+}$ concentration increased. Only in RT4 spheroids did an increase in $Ca^{2+}$ from 2 to 5 mM slightly enhance YP fluorescence intensity. In these spheroids, exposure to nsPEFs at 5 mM $Ca^{2+}$ also resulted in a smaller increase in the 2D projected cell area compared with 2 mM $Ca^{2+}$. Thus, reduced swelling in RT4 spheroids may have confined fluorescence signals to a smaller area, leading to an apparent increase in mean fluorescence intensity within the 2D projection.

Disassembly of actin structures and loss of cellular stiffness are known secondary effects of cell swelling induced by membrane permeabilization following nsPEF exposure[42]. In this study, we demonstrated that the extent of spheroid stiffness loss corresponds to the differential severity of membrane disruption, with spheroids derived from normal SV-HUC-1 cells being less affected than those from malignant RT4 cells. Accordingly, urothelial cancer cells exhibited not only higher YP uptake but also more pronounced immediate secondary effects of membrane permeabilization.

Although plasma membrane permeabilization in our study was achieved using nsPEFs, effective membrane disruption can also be induced by chemical agents or membrane-permeabilizing proteins delivered via intravesical instillation. Such approaches may offer therapeutic potential for treating broader or flat tumor regions, such as carcinoma in situ of the urothelium. High-grade tumors typically lack a well-differentiated umbrella cell layer and show alterations in the protective glycosaminoglycan coating, potentially making these cells more susceptible to agents delivered intravesical[56,57]. A key advantage of nsPEFs, however, lies in their ability to permeabilize not only superficial lesions but also deeper, infiltrating tumor regions that are otherwise less accessible to intravesical therapies.

The observed differences in membrane durability and repair dynamics between urothelial cancer and normal cells provide a promising foundation for the development of selective treatment strategies. A deeper understanding of membrane susceptibility and the underlying repair mechanisms will be essential for advancing membrane-targeted therapies and improving oncological outcomes.

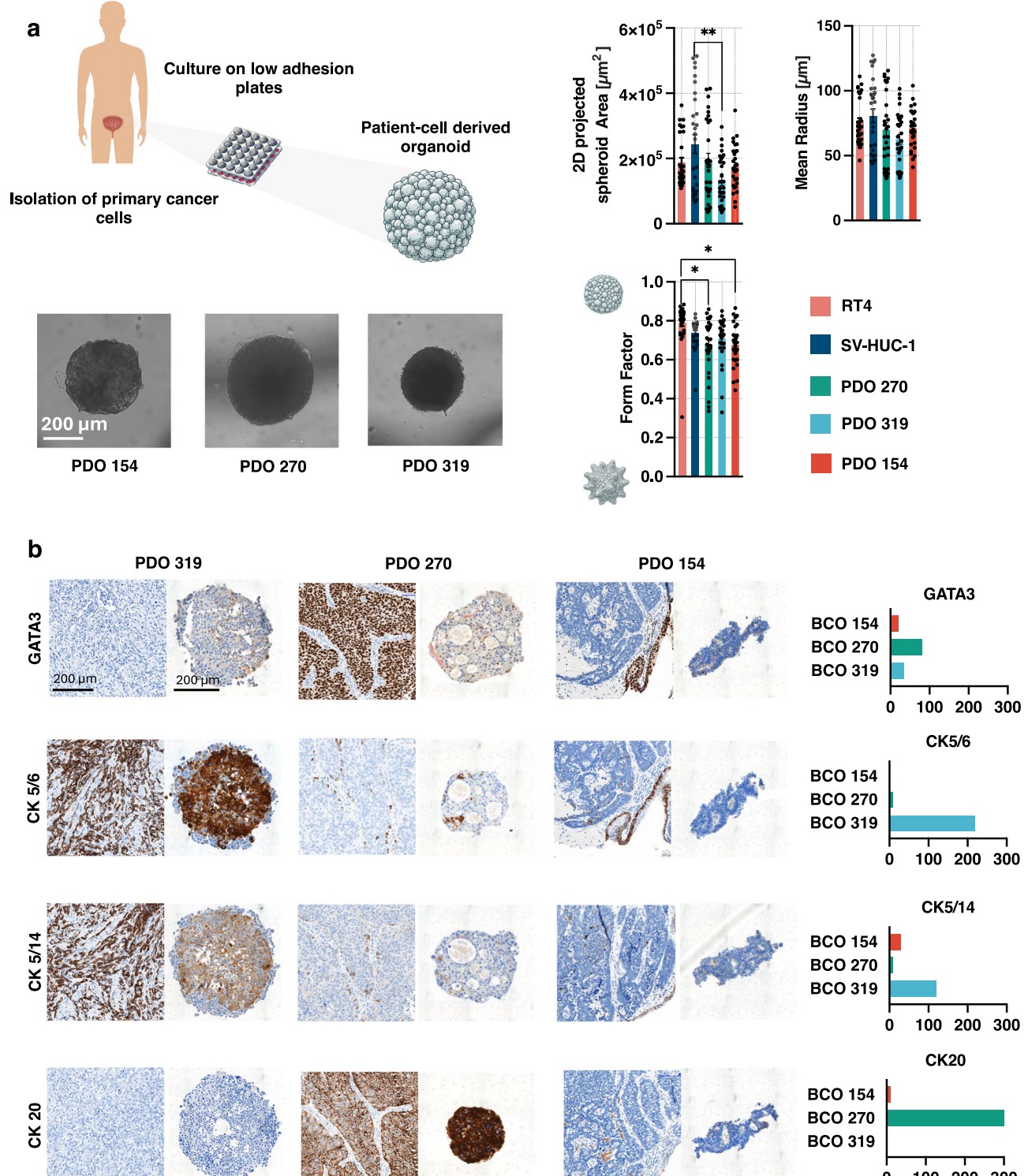

**Fig. 4 | Characterization of Patient-Derived Organoids (PDOs). a** Representative image of PDOs and Bar graphs showing the distribution of the 2D projected cross-sectional area, mean radius, and form factor of PDOs from primary urothelial cancer cells obtained from patients with bladder cancer (mean ± SEM; $n$ = 30). Statistical comparisons between Spheroids and PDOs were performed using Welch's ANOVA, followed by Dunnett's T3 post hoc testing. Welch's ANOVA revealed only minor but statistically significant differences in the average 2D projected spheroid area (W(4.000, 71.31) = 4.286) and form factor (W(4.000, 123.2) = 5.737), while no significant difference was observed in the mean radius (W(4.000, 71.33) = 2.390) between PDOs and spheroids. **b** Representative histological images of spheroid cross-sections and their corresponding primary tumor tissues are shown, stained with immunohistochemical markers GATA3, CK5/6, CK5/14, and CK20. Accompanying histoscore bar graphs display the pathologist-assigned histoscores for the main urothelial markers in the spheroids.

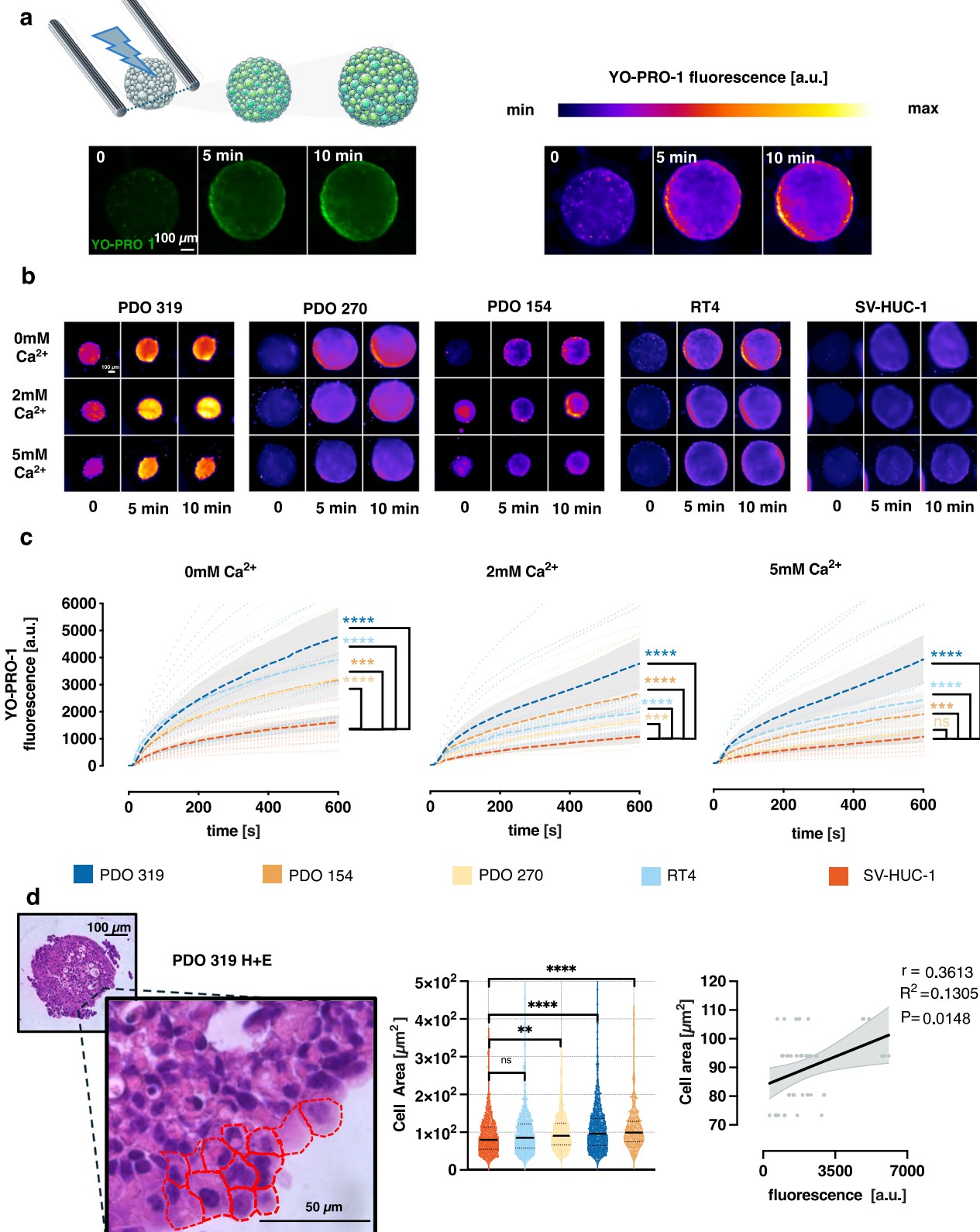

## Material and methods

### Cell line culture

Experiments were conducted on T24, UM-UC-3, RT4 human cancer urothelial cells and HBLAK and SV-HUC-1 normal human urothelial cells. T24, UM-UC-3, RT4 and SV-HUC-1 cell line were acquired from American Type Culture Collection (ATCC, Manassas, VA, USA). HBLAK cell line was acquired from CELLnTEC (CELLnTEC Advanced Cell Systems, Bern, Switzerland). Cells were cultured at 37 °C in a 5% $CO_2$ atmosphere in their recommended media and expanded to 80–90% confluency. All media were supplemented with 2 mM L-Glutamine, 10% FBS, and 1X Penicillin-Streptomycin (Gibco, Paisley, UK). Cell line identity was verified after acquisition based on morphology and growth characteristics and regularly

**Fig. 5 | Integration of PDO and spheroid models in electroporation studies.**
**a** Spheroids and PDOs were positioned between two stainless steel electrodes. YP uptake was monitored via fluorescence imaging for 10 min post-exposure to nsPEFs. Fluorescence images were pseudocolored to enhance visualization. **b** Representative pseudocolored images of spheroids and PDOs before and after nsPEFs exposure. **c** Time-course analysis of YP fluorescence uptake in spheroids derived from normal urothelial cells and urothelial cancer cells, under different extracellular $Ca^{2+}$ concentrations. The spheroids were observed for 10 min following exposure. Data were shown as mean ± SEM ($n$ = 9–11). Statistical comparisons of the AUC of fluorescence between normal urothelial SV-HUC-1 spheroids and cancer RT4 spheroids with PDO were performed using Welch's ANOVA, followed by Dunnett's T3 post

hoc test for multiple comparisons: for 0 mM $Ca^{2+}$, $W(4, 28.14) = 28.34$, for 2 mM $Ca^{2+}$, $W(4, 20.46) = 38.53$ and for 5 mM $Ca^{2+}$, $W(4, 18.09) = 53.60$. **d** Quantification and correlation of single-cell size in PDOs and YP fluorescence after exposure. The mean cell size was calculated based on H&E-stained slides of spheroids. Statistical comparisons of single cell area between spheroids and PDOs were performed using Welch's ANOVA, followed by Dunnett's T3 post hoc test for multiple comparisons ($W(4, 787.4) = 10.74$). The correlation between mean single-cell size in spheroids and YO-PRO-1 fluorescence intensity 10 min after nsPEF exposure in 2 mM $Ca^{2+}$ was assessed by linear regression analysis. The analysis included data from 45 spheroids and PDOs derived from RT4, SV-HUC-1, PDO 154, PDO 270, and PDO 319. Each dot represents a single spheroid ($n$ = 45).

confirmed against reference images and literature descriptions. All cultures were routinely tested and confirmed negative for mycoplasma contamination.

In this study, we refer to 3D cultures derived from primary urothelial cells as "organoids" and those derived from established cell lines as "spheroids". To generate spheroids, RT4 and SV-HUC-1 cells were seeded into 96-well plates pre-coated with 1% agarose to create a low-attachment environment. For coating, 1% (w/v) agarose solution (Merck KGaA, Darmstadt, Germany) was prepared in deionized water and autoclaved. After brief heating of the solution in a microwave for one minute, 75 µL was dispensed into each well to, form a meniscus at the bottom. The plate was left at room temperature for 10 min to allow the agarose to cool and solidify, creating non-adherent, round-bottom wells. The cells were then added at the desired density in a final volume of 150 µL per well. Half of the medium was replaced every 2–3 days. Cultures were maintained for up to 3 weeks to allow spheroids to reach the appropriate size for experimental use. Once the spheroids acquired a uniform, spherical morphology, they were gently collected and transferred onto glass slides using a standard 1000 µL pipette in preparation for the electroporation procedure.

### PDOs culture
After informed consent of the patient, tumor cells were retrieved from surgical specimens of a radical cystectomy or transurethral bladder tumor resection (see supplementary Table S1 for clinical data of patients). This study was approved by the Scientific Board at the University of Tuebingen (804/2020/B02). All experiments with the samples were performed in accordance with German regulations and ethical standards as laid down in the Declaration of Helsinki. Pathologist confirmed urothelial carcinoma samples were obtained from patients undergoing surgery after written and informed consent. All ethical regulations relevant to human research participants were followed. Subsequently the tissue was minced and collected by centrifugation (480 g, 10 min. ambient temperature). The sediment was resuspended in buffer containing collagenase (3000 U/mL) and hyaluronidase (1000 U/mL) (STEMCELL Technologies, Vancouver, Canada), and incubated under moderate agitation (37 °C, 30 min). The proteolytic degradation was continued by adding fresh collagenase for 30 min at 37 °C. Debris was removed using a cell strainer (70 µm mesh) and the filtrate was sedimented by centrifugation (150 g, 7 min, room temperature). Cells were resuspended in 10 µL and further mixed on ice with 30 µL BME (BioTechne, Minneapolis, MN, USA) and dipped in a 24-well plate and flipped upside-down to generate hanging drops. After incubation at 37 °C for 30 min, the plates were turned back, complemented with 500 µL bladder tumor medium (BTM) culture medium (50 ml BTM containing: 22.5 ml L-RN-Media, 22 ml advanced DMEM (1X)/F-12, 2.5 ml csFBS, 1 ml B27 supplement, 500 µl L-glutamine, 500 µl 1 M HEPES, 500 µl 1 M nicotinamide, 125 µl 500 mM N-acetylcysteine, 50 µl 5 mM A83-01, 50 µl 50 mg/ml Primocine, 50 µl 100 µg/ml FGF-10, 25 µl 50 µg/ml FGF-7, 12.5 µl 50 µg/ml FGF-2, 5 µl 100 mM Y-27632, 0.5 µl 500 µM EGF), per well, and incubated in a cell-culture incubator (37 °C, 5% $CO_2$, humidified atmosphere). To ensure genotypic and phenotypic stability, the primary cell lines were passaged four times before use and employed for experiments up to passage eight. Cell plates containing primary cells in Matrigel were placed on ice to liquefy the

Matrigel and collected with ice cold phosphate-buffered saline (PBS). The cell suspension in Matrigel was then centrifuged and resuspended in BTM culture medium and dipped in a new 24-well plate.

To generate spherical PDOs, primary urothelial cells were seeded into 96-well plates pre-coated with 1% agarose and cultured in BTM medium, following a protocol analogous to that used for cell line-derived spheroids. Upon reaching the appropriate size for experimental use, the PDOs exhibited strong structural integrity, allowing for easy collection using a standard 1000-µL pipette and enabling their application in subsequent electroporation studies.

### Immunohistohemical characterization of PDOs and tissue samples
PDOs were rinsed twice with PBS. PDOs were fixed by 4% formaldehyde (30 min, a.t.), Subsequently PDOs were dehydrated and embedded in paraffin. 3 µm Paraffin sections were generated using Leica RM2125 microtome (Leica, Nussloch, Germany) dewaxed and rehydrated by aid of incubation in xylene, rehydrated by ethanol in decreasing concentrations, and mounted on glass slides. Immunohistochemistry of paraffin sections was utilized to detect expression of the bladder cancer stem-cell markers in PDOs and bladder cancer tissue samples from the patients corresponding to the PDO 270, PDO 319, PDO 154. The samples were counterstained by HE, covered (VectaMount, Vectorlabs), and recorded by microscopy DMI8 inverted microscope (Leica Microsystems, Wetzlar, Germany).

The immunohisochemical characterization o PDO was performed in an immunohistochemistry laboratory with accreditation by the German Accreditation Office (DAKKs) using an automated stainer (Ventana Medical Systems, Tucson, Arizona, USA) in accordance with the manufacturer's protocol. The following antibodies were used for analysis: Cytokeratin 20 (CK20, Ks20.8, DAKO (Glostrup, Denmark) 1:400), Cytokeratin 5/14 (CK5/14, XM26 & LL002, Zytomed Systems (Berlin, Germany) 1:200), Cytokeratin 5/6 (CK5/6, D5/16B4, DAKO, 1:150), GATA3 (l50-823, Roche Diagnostics (Rotkreuz, Switzerland, ready-to-use). The expression of the four markers was evaluated by an experienced pathologist and categorized as strong, moderate, or weak. An H-score was then calculated using the formula: H-score = $[(0 \times \%$ negative cells$) + (1 \times \%$ weakly positive cells$) + (2 \times \%$ moderately positive cells$) + (3 \times \%$ strongly positive cells$)]$, resulting in a total score ranging from 0 (no staining) to 300 (uniform strong staining in all cells).

### Solutions and sample preparation
The physiological solution used for electroporation studies was formulated to closely mimic the ionic composition, pH, osmolality, and conductivity of the extracellular environment. It contained 140 mM NaCl, 5.4 mM KCl, 2 mM $MgCl_2$, 10 mM glucose, and 10 mM HEPES, with an osmolality of 300 mOsm/kg, and a conductivity of 1.4 S/m. To investigate the role of $Ca^{2+}$ in membrane repair, three versions of the solution were prepared, containing 0 mM, 2 mM, and 5 mM $CaCl_2$. The pH of both solutions was adjusted to physiological pH of 7.2–7.4 with 5 M NaOH. Fluids underwent vacuum-driven filtration with Steriflip (Millipore, Darmstadt, Germany) and have been used for experiments within two weeks.

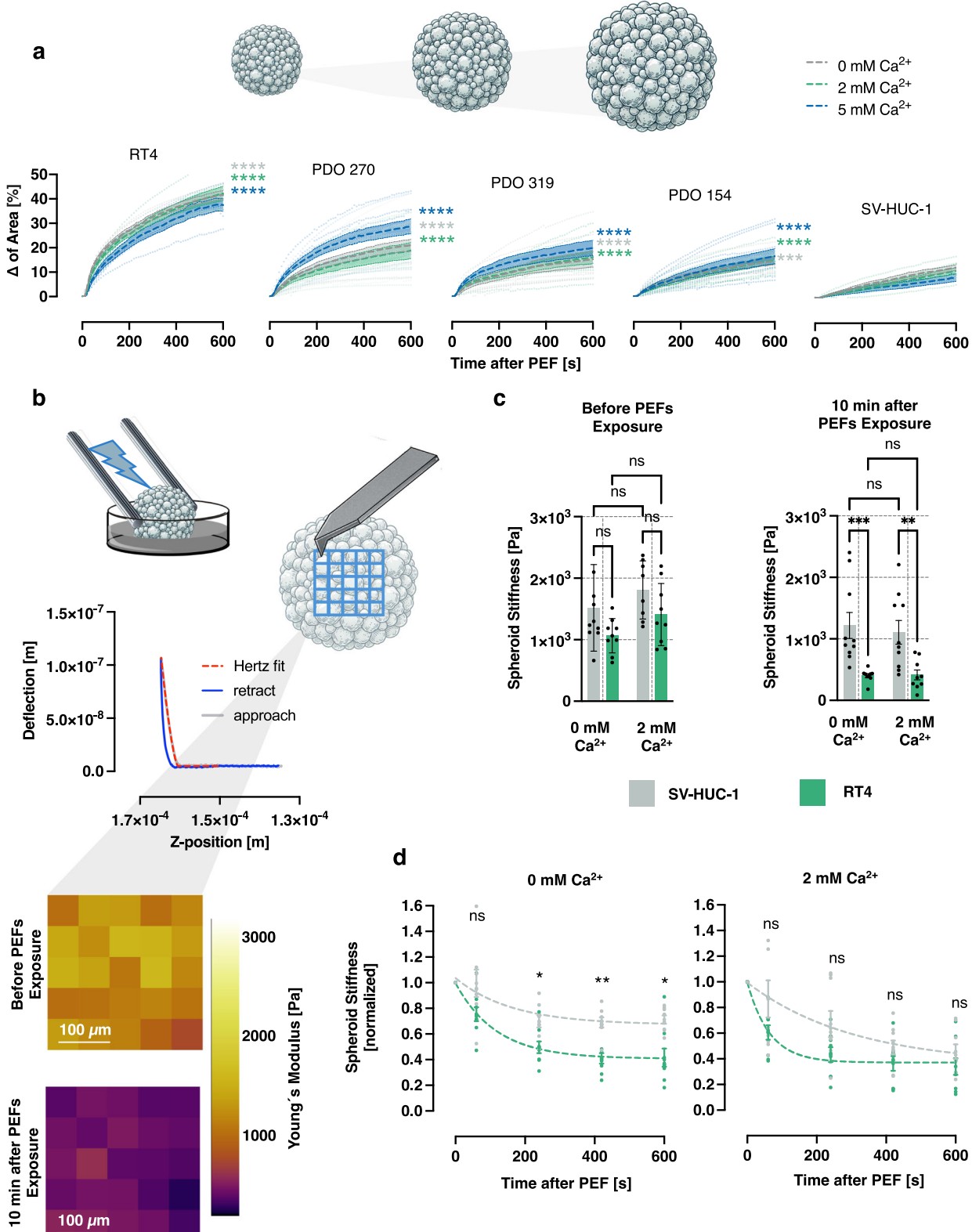

For electroporation of cell monolayers, cells were seeded onto 25 mm glass coverslips (Carl Roth, Karlsruhe, Germany) at a density of $0.15 \times 10^6$ cells per coverslip in DMEM (PAN-Biotech Aidenbach, Germany) supplemented with 10% fetal bovine serum (FBS) (PAN-Biotech). After 24 h of incubation, a homogeneous monolayer was formed. Prior to the experiment, the culture medium was removed, and the monolayers were rinsed twice with a physiological solution lacking $Ca^{2+}$. Cells were then incubated in physiological solution containing 0 mM, 2 mM, or 5 mM $Ca^{2+}$, supplemented with 1 μM YP dye (Thermo Fisher Scientific, Waltham, MA, USA) nucleic acid stain, for 2 min before nsPEFs exposure.

For electroporation of spheroids or PDOs, samples were initially placed on a sterile microscope glass slide and gently rinsed three times with a

**Fig. 6 | Evaluation of spheroid swelling and stiffness loss after electroporation.** **a** Changes in the 2D projected area of spheroids and PDOs (mean ± SEM, $n$ = 9–11) following nsPEFs exposure in solutions containing 0 mM, 2 mM, or 5 mM Ca²⁺. Statistical comparisons of the area under the curve (AUC) for 2D projected area changes over time were performed using Welch's ANOVA, followed by Dunnett's T3 post hoc test to compare normal urothelial SV-HUC-1 spheroids with cancer RT4 spheroids and PDOs in different Ca²⁺ concentrations: for 0 mM Ca²⁺, $W(4, 21.73) = 835.0$; for 2 mM Ca²⁺, $W(4, 19.38) = 485.5$; and for 5 mM Ca²⁺, $W(4, 19.46) = 326.2$. **b** Mechanical properties of spheroids of normal SV-HUC-1 and cancer RT4 cells were evaluated using AFM force mapping, conducted at 25 points $(5 \times 5$ grid) across the spheroid surface at defined time intervals post-exposure. Young's modulus values were calculated from force–distance curves by fitting them to the Hertz model. Values obtained from a single spheroid were averaged to yield one representative value per spheroid. **c** Comparison of Young's modulus ("stiffness") between SV-HUC-1 and RT4 spheroids in 0 mM and 2 mM Ca²⁺ solutions,

both before and 10 minutes after exposure to nsPEFs ($n$ = 9–11). Data are presented as mean ± SEM. Statistical comparisons were performed using two-way ANOVA with Geisser–Greenhouse correction for unequal variability: Before exposure: effect of Ca²⁺, $F(1, 32) = 3.315$; effect of cell line, $F(1, 32) = 5.963$. After exposure: effect of Ca²⁺, $F(1, 34) = 0.104$; effect of cell line, $F(1, 34) = 23.24$. **d** Young's modulus values were normalized to the pre-exposure baseline and the mean values were fitted using a single-exponential nonlinear regression model (mean ± SEM, $n$ = 9–11): R² for the exponential fits were 0.85 and 0.99 for SV-HUC-1 spheroids, and 0.95 and 0.94 for RT4 spheroids under 0 mM and 2 mM Ca²⁺ conditions, respectively. Points represent individual measurements from independent experimental replicates. For statistical comparisons at defined time points, repeated measures two-way ANOVA with Geisser–Greenhouse correction was applied, followed by Šidák's multiple comparisons test. In 0 mM Ca²⁺: effect of time, $F(2.235, 33.53) = 48.26$; effect of cell type, $F(1, 15) = 15.22$. In 2 mM Ca²⁺: effect of time, $F(1.665, 26.65) = 55.19$; effect of cell type, $F(1, 16) = 3.252$.

physiological solution lacking Ca²⁺. They were then transferred into a low-profile Glass Bottom μ-Dish, 35 mm (Ibidi, Gräfelfing, Germany), and fully immersed in physiological solution containing 1 μM YP. The dish was placed on the microscope stage, and spheroids were allowed to settle for 5 min to ensure contact with the bottom surface. Once settled, the spheroids remained stable, enabling precise micromanipulation and controlled exposure to pulsed electric fields.

### Electrode arrays and electroporation procedure
Array of two tungsten rods (0.5 mm diameter) were isolated and connected in 1.5 mm center-to-center spacing. A custom-made 3D printed electrode holder positioned the tungsten electrodes touching the coverslip at a stable 45° angle between coverslip and electrodes[58]. The contact electrodes were designed to produce an electric field that gradually decayed with distance from them, enabling the comparison of cell killing across a range of electric field strengths within a single sample. In this study, we used a custom-built pulse generator developed at the Institute of High Magnetic Fields (VGTU, Vilnius, Lithuania), capable of delivering electric pulses ranging from 100 nanoseconds (ns) to 1 millisecond (ms)[59]. For nsPEFs exposure, trains of 200, 300-ns nearly rectangular pulses at 1 kV were delivered at 10 Hz. Electric field distribution was simulated using Sim4Lifelite v7.3.0 software (Zurich Med Tech, Switzerland), matching experimental electrode positions. Electric field values were calculated at a plane 5 μm above the well bottom for 1 V applied between electrodes and then scaled to the applied voltage (see supplementary Fig. S1a–e for details). Pulse shape and amplitude were monitored with a TBS1052C oscilloscope (Tektronix, Beaverton, OR, US). Electric field parameters—pulse number, frequency, and intensity—were optimized to cause the electroporation of different cell lines and PDOs in region between the electrodes.

In monolayer experiments, groups of over 200 cells were simultaneously exposed to nsPEFs across a broad range of electric field intensities. In contrast, for experiments involving PDOs or spheroids, only one spheroid was exposed to nsPEFs at a time. Comparative experiments examining different cell lines and varying Ca²⁺ concentrations were conducted side-by-side and randomized to minimize bias.

For fluorescence imaging, we utilized a Leica DMI8 inverted microscope (Leica Microsystems, Wetzlar, Germany) equipped with a Leica K3C KIT camera (Leica Microsystems, Wetzlar, Germany) and pE-300 lite icool led illuminator (CoolLED Ltd., Hampshire, United Kingdom). YP fluorescence in selected groups of cells was monitored by time-lapse imaging using a standard FITC filter cube and a 10× dry objective.

Changes in YP fluorescence are commonly used as a semi-quantitative or quantitative indicator of plasma membrane permeabilization[28,60,61]. For analysis of cell monolayers, the focal plane was set during the initial image acquisition and maintained unchanged throughout the entire time-lapse imaging sequence. Images were captured every 4 seconds (s) over a 3-minutes (min) period. To quantify membrane permeabilization, baseline YP fluorescence recorded prior to nsPEF exposure was subtracted from the

average fluorescence intensity measured at each subsequent time point (see supplementary Fig. S9).

For spheroid imaging, the camera was focused on focus on the largest cross-sectional area of the spheroid and images were acquired every 10 s for a total duration of 10 min. All experimental sets included sham-exposed controls (no electroporation) to ensure that any increase in YP fluorescence was not due to cell degradation over time, particularly in spheroid samples (See supplementary Figs. S2a and S7a for the results of control exposures). Illumination intensity and camera sensitivity were adjusted to ensure reliable detection of YP uptake while avoiding camera saturation and minimizing photobleaching. These settings remained constant within each experimental set. The fluorescence intensity values are reported in arbitrary units (a.u.) and are consistent within each figure.

### Assessment of cellular morphology and size in cultured cells and tissue sections
For analysis of cell morphology, cells were seeded onto Ibidi μ-Plate 24 black well plate (Ibidi, Gräfelfing, Germany) and stained with Alexa 546 Phalloidin (Thermo Fisher Scientific) to visualize F-actin and Hoechst 33342 (Thermo Fisher Scientific) for nuclear staining, following the standard staining protocol. Fluorescence images were acquired using a Leica DMI8 inverted microscope. The parameters of cell morphology have been determined in all cells within two randomly chosen 250 μm × 250 μm region of interest (ROI) in three different wells. Image analysis was conducted using CellProfiler software (version 4.2.6)[62]. The cell area was determined by segmenting phalloidin-stained cytoskeleton and linking it to the corresponding nuclei identified by Hoechst staining. After cell segmentation, CellProfiler calculated the form factor for each identified object based on its shape parameters (form factor = $4\pi \times$ (area × perimeter$^{-2}$)). This value ranges from 0 to 1, where 1 represents a perfect circle and lower values indicate increasingly elongated or irregular shapes.

H&E staining was performed on tissue microarray (TMA) sections containing cores from primary urothelial tumors, lymph node metastases, and normal urothelium. TMAs were constructed using a tissue array (Beecher Instruments, Silver Springs, MD, USA) as previously described[63]. Representative tumor and normal regions were identified microscopically, and two 0.6 mm-diameter cores were collected from each area to account for intratumoral heterogeneity.

TMA slides were imaged using Pannoramic MIDI II digital slide scanner (3DHISTECH Ltd., Budapest, Hungary). Cancer and normal tissue regions were annotated and verified under the supervision of a trained pathologist. Following tissue classification, single-cell detection and quantification were performed in QuPath (version 0.5.1) using brightfield images stained with H&E[64]. Within defined tissue compartments, nuclei were detected using QuPath's built-in Cell Detection algorithm, which identifies nuclei based on hematoxylin staining intensity (see supplementary Fig. S4 for details). The algorithm applies color deconvolution to separate hematoxylin and eosin channels, allowing robust nuclear detection based on the

hematoxylin signal. For each detected cell, morphometric and features including cell area and form factor were extracted.

## Analysis of YP fluorescence changes in cell monolayer

For YP uptake analysis an $800 \times 600$ µm rectangular ROI was selected. The ROI was positioned such that one of its 800 µm long edges overlapped the line connecting the tips of the two electrodes, with each corner of this edge offset by 100 µm from the respective electrode tip. This placement positioned the entire $800 \times 600$ µm rectangular ROI just beyond the electrode gap, on the side distal to the electrodes (see supplementary Fig. S1a, b and Fig. S10a, b for visualization). This placement ensured that the ROI encompassed the area of interest directly affected by the electric field, while avoiding the regions of strongest field gradients immediately adjacent to the electrodes. ROI was divided into three equal subregions (subROI), each measuring $800 \times 200$ µm, by drawing two parallel lines, evenly spaced along the shorter axis of the ROI. The variation in electric field strength within individual sub-ROI—between the cells nearest and farthest from the electrodes—was less than 20%, ensuring relatively homogeneous exposure conditions (see supplementary Fig. S1b for details).

In subsequent evaluations, cell area segmentation in brightfield images was performed using a combined approach involving CellProfiler and Ilastik software[65,66]. Ilastik employs machine learning-based pixel classification. Initially, regions of interest—such as cell areas and background—were manually annotated to create a training dataset. These annotations were used to train a supervised machine learning model that classifies pixels based on intensity, texture, and edge features. The software then generates a grayscale probability map by assigning each pixel a likelihood of belonging to a specific class. After training on representative images from multiple cell lines, Ilastik was used to generate probability maps for each frame in the bright-field time-lapse imaging sequence. This ensured that dynamic changes in cell size, including osmotic swelling after electroporation, were considered in the definition of cell borders, allowing accurate and adaptive boundary segmentation throughout the time-lapse sequence. The resulting probability maps were imported into CellProfiler, where they were used to quantify changes in fluorescence within regions identified as cell-covered, based on Ilastik-guided segmentation (see Supplementary Fig. S10 for visualization).

## Analysis of YP fluorescence changes and the size changes of Cell Spheroid SuSand PDOs

For spheroids and PDOs, the same Ilastik-based segmentation strategy was applied. Grayscale probability maps were generated for each frame in the time-lapse sequence, allowing precise delineation of the organoid boundaries over time. These maps were subsequently analyzed in CellProfiler, enabling accurate quantification of both dynamic changes in organoid size and mean YP fluorescence intensity following exposure to pulsed electric fields (see supplementary Fig. S6 for visualization).

## Stiffness measurement of cell spheroid

For stiffness measurements, spheroids derived from SV-HUC-1 and RT4 cells were seeded onto tissue culture dishes with a glass-bottom coverslip (FD5040-100, World Precision Instruments, Sarasota, FL) and incubated overnight in cell culture medium to ensure stable adhesion to the substrate. 5 min before the atomic force microscopy (AFM) experiment, the cell culture medium was replaced with the physiological solution supplemented with either 0 mM or 2 mM $Ca^{2+}$. AFM was performed on a custom-built setup using cantilevers with a sphere tip (biosphere B1000-CONT, Nanotools GmbH, Munich, Germany). The spring constant of each cantilever was calibrated beforehand using the thermal noise method, giving about 0.04 N/m[67]. A force map consisting of 25 force curves ($5 \times 5$ grid) were acquired on one spheroid for each time point after exposure in each condition. The data were analyzed using Igor Pro 9 (WaveMetrics Inc, Lake Oswego, OR). The Young's modulus ("stiffness") was obtained for each force curve by fitting the Hertz model to the approach part of the curve.

## Statistics and reproducibility

Fluorescence measurements of YP uptake of single cells were averaged for one sub-ROI in each timepoint. Each experiment was repeated three times independently (biological replicates), with 6–7 technical replicates per condition in each experiment. All technical replicates were treated as independent data points for statistical analysis, resulting in a total of $n = 18$–20 per condition.

Experiments on spheroids were conducted in three independent biological replicates, with 3–4 spheroids analyzed per condition in each replicate. Each spheroid was treated as an independent data point, resulting in a total of $n = 9$–11 per condition

The Brown–Forsythe test was used to assess equality of variances, with $p < 0.05$ indicating that the variances were significantly different.

To quantitatively compare YP fluorescence uptake as well as changes in the 2D projected area of spheroids and PDOs over time following exposure the area under the curve (AUC) was calculated for each replicate, representing the cumulative fluorescence signal of size change over time. Group differences in AUC values were assessed using Welch's ANOVA, followed by Dunnett's T3 post hoc testing to account for multiple comparisons in the presence of unequal variances. All data are reported as mean ± SEM, with statistical significance defined as $p < 0.05$. W statistic, and degrees of freedom are reported in figure legends.

A statistical comparison of fluorescence intensity between different cell lines or spheroids and PDOs at the final time point of observation, was also performed using Welch's ANOVA, followed by Dunnett's T3 post hoc testing.

YP fluorescence uptake over time was analyzed by fitting the average response for each experimental condition to a one-phase exponential model. The fits showed a strong correlation, with coefficients of determination ($R^2$) exceeding 0.95 for all cell lines under all tested conditions, indicating robust exponential uptake kinetics. Time constants ($\tau$), representing the dynamics of fluorescence progression, were extracted from the best-fit curves generated from individual experimental replicates. The average time constants for each condition are presented as mean ± SEM. Differences in time constants across various $Ca^{2+}$ concentrations for each cell line were assessed using Welch's ANOVA, followed by Dunnett's T3 post hoc testing to correct for multiple comparisons. W statistic, and degrees of freedom are reported in figure legends.

Cell morphology parameters after staining with phalloidin were quantified in two randomly selected ROIs within each of three wells. This analysis was conducted across three biological replicates to account for potential variability related to cell passage. The average cell size within each ROI was treated as an independent data point for statistical analysis, resulting in $n = 18$. Data were visualized using violin plots, and statistical comparisons between cell lines were performed using Welch's ANOVA, followed by Dunnett's T3 post hoc testing to account for multiple comparisons in the presence of unequal variances.

For tissue samples, the average cell area as well as cell form factor was calculated across multiple specimens of normal urothelium, primary tumors, and lymph node metastases, using 9 tissue microarrays (TMAs). In total, the analysis included 47 samples of normal urothelial tissue, 66 samples of urothelial cancer from primary tumors, and 56 samples from lymph node metastases. Statistical comparison of mean urothelial cell area and form factor across normal urothelial tissue, urothelial carcinoma, and lymph node metastasis was performed using weighted one-way ($n = 9$), with the number of tissue sections per array used as weights.

The Young's modulus ("stiffness") of a single spheroid was obtained as the median of one force map ($5 \times 5$ force curves) to yield one representative value per spheroid. The experiment was conducted with 3–4 technical replicates across 3 independent biological replicates. Each spheroid was treated as a single data point, resulting in a total of $n = 9$–10. Data are presented as mean ± SE.

Statistical comparisons of normal urothelial SV-HUC-1 and urothelial cancer RT4 spheroid stiffness in 0 mM and 2 mM $Ca^{2+}$ physiological solutions for both before and after exposure to nsPEFs, were performed using

two-way ANOVA with Geisser–Greenhouse correction for unequal variability. To analyze temporal changes in stiffness, Young's modulus values were normalized to the pre-exposure baseline and fitted using a single-exponential nonlinear regression model. For statistical comparisons at defined time points, repeated measures two-way ANOVA with Geisser–Greenhouse correction was applied, followed by Šidák's multiple comparisons test.

Data fits and graphs were prepared with GraphPad Prism Software (Version 10.1.1. GraphPad Software, San Diego, CA, USA). Asterisks in all figures indicate levels of statistical significance as follows: $p \leq 0.05$ (*), $p \leq 0.01$ (**), $p \leq 0.001$ (***), and $p \leq 0.0001$ (****); ns indicates not significant ($p > 0.05$).

### Reporting Summary

Further information on research design is available in the Nature Portfolio Reporting Summary linked to this article.

### Data availability

The supporting numerical source data for this study are provided in the Supplementary Data. Correspondence and requests for materials should be addressed to Aleksander Kielbik.

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

## Acknowledgements

This work was conducted in the framework of the Graduate School 2543/1 "Intraoperative Multisensory Tissue Differentiation in Oncology" (project ID 40947457) funded by the German Research Foundation (DFG). We thank Prof. Dr. W.K. Aicher for his valuable scientific input and experimental support throughout the project.

## Author contributions

Conceptualization: A.K. Methodology: A.K., P.S., V.B., E.H., and V.N. Validation: V.B., F.F., T.E.S., B.A., and I.T. Formal Analysis: A.K., E.H., and V.B., Investigation: A.K, E.H., D.L, M.K., P.S., V.B., H.P., and S.W. Resources: B.A., O.V., M.L.B., S.W., T.E.S., and F.F. Data Curation: A.K. Writing—Original, Draft: A.K Writing—Review & Editing: A.K. and P.S., Visualization: A.K. Supervision: F.F., T.E.S., M.L.B., O.V., and I.T. Project Administration: M.L.B., O.V., I.T., and B.A., Funding Acquisition: M.L.B., O.V., I.T., B.A, T.E.S., and F.F. All authors reviewed, edited, and approved the final version of the manuscript.

## Funding

## Competing interests

The authors declare no competing interests.
