## [Transparent Peer Review file · Communications Biology]

Nanosecond pulsed electric fields induce cell-size-dependent selective permeabilization of urothelial cancer cells

Corresponding Author: Dr Aleksander Kielbik

Version 0:

Reviewer comments:

Reviewer #1

(Remarks to the Author)

This study describes an attempt to develop a targeted cancer therapy method using a nanosecond pulsed electric field (nsPEF) to induce membrane permeability. Using a fluorescence dye uptake assay with two-dimensional cultured urothelial cells, the authors demonstrated that nsPEF-induced membrane permeability is more selective toward cancer cells than toward normal cells. Furthermore, patient-derived organoids were used to show that nsPEFs induce significant swelling and reduced stiffness in cancer cells. The presented data suggest the therapeutic potential of nsPEFs for treating malignant urothelial cancer.

This study is technically sound, and the presented data are compelling. However, several concerns remain, such as the use of different cell lines in two-dimensional (2D) culture and organoid formation, as well as the lack of direct comparisons between experimental conditions. Furthermore, the relationship between the results obtained from organoids and the claims made in this paper is unclear.

Major points:

1. In the 2D culture experiment, the authors found a correlation between cell size and YP uptake. In the experiment using patient-derived organoids, discussions regarding organoid size were presented; however, the size of the cells that comprise the organoids was not investigated. Because different types of urothelial cancer cells were used in the 2D and organoid cultures, the results from each experiment cannot be compared. Additionally, it should be noted that the 2D projected area and form factor of organoids differ from the morphological parameters of their constituent cells.
2. Figure 5d shows that YP uptake into RT4 cells increases with 5 mM Ca^{2+} compared to 2 mM. This behavior differs from that of other cells, and possible reasons should be discussed.
3. Figure 6a shows that urothelial cancer-derived organoids swell when treated with nsPEFs. Figures 6b–d also demonstrate that nsPEFs decrease organoid stiffness. These results reflect the degree of membrane permeabilization caused by nsPEFs and align with the YP uptake results presented in Figure 5. However, this paper does not discuss the therapeutic significance of cellular swelling or stiffness reduction. Therefore, the significance of the data presented in Figure 6 is unclear.
4. Lines 407–408 state that extracellular Ca^{2+} enhances the reduction in spheroid stiffness. However, Figure 6c shows that there was no significant difference between 0 mM and 2 mM Ca^{2+} in normal or cancer cells. Please explain the validity of this interpretation.
5. The graph in Figure 5c does not appear to correspond to the YP uptake fluorescence images in Figure 5b. For instance, PDO270 seems to have the second-highest YP uptake after 319, but the graph always follows the trajectory closest to SV-HUC-1 (normal cell). Additionally, it seems that the colors in the graph legend are reversed for PDO154 and PDO270. Please confirm.

Minor points:

1. In Figure 2b, the colors in the graph legend are reversed for 0 mM Ca^{2+} and 10 kV/cm, as well as for 2 mM Ca^{2+} and 10 kV/cm. This needs to be corrected.
2. In line 289, "(Figure 5b)" should be "(Figure 5c)."
3. In Figure 6d, "(sek)" should be "(sec)."
4. Line 94: Since the term "AUC" appears for the first time in the main text, the full spelling should be provided.
5. Lines 288–304, 311–316, and 340–363 offer thorough explanations of the numerical values depicted in the graphs.

However, these explanations are overly verbose. To improve readability, please revise the text to provide brief summaries of the key points.

Reviewer #2

(Remarks to the Author)

COMMENTS FOR THE AUTHORS

I found the enclosed manuscript COMMSBIO-25-6484-T entitled "Exposure to Nanosecond Pulsed Electric Fields Reveals Increased Cell Membrane Vulnerability and Impaired Cell Membrane Repair in Urothelial Cancer Cells." by Kielbik et al. suitable for publication in Communications Biology after minor corrections are made.

Electroporation have been successfully used in many applications in medicine, such as treating several types of cancer, drug delivery, gene therapy and cardiac arrhythmia. Therefore, exploring new possibilities of medical treatments is of significant importance. In this research, the authors study the mechanisms of electroporation effects on normal and cancer urothelial cells for developing treatment of urinary bladder cancer. The study is very well designed and executed, with great care for details. The statistical analysis is appropriate. The results are efficiently presented and discussed. There are only minor issues to be resolved, mainly in the presentation of the results.

Here are my comments and suggestions:

1. Results (line 94): Define the abbreviation AUC on the first appearance.
2. Supplement, Figures S2 and S3: From Figures S2b and S3 (the first in a and last in b) it can be concluded that images (the modelled one and the microscopic one) are in the opposite direction. It would be better to match the model with the experimental image. Moreover, if the sub-ROI in S3b match the regions of E field in S2b, different E should also be noted in S3b (last image) for greater clarity. The same goes for Figures 1a, b.
3. Figure 1b: It is not clear at which axis the graph of E was determined. Why is half of it black and half of it green. What does it mean "across the region of interest"?
4. Figure 2a: What is the cell extension on the second image of the schematic?
5. Figure caption 2 (line 954): Considering the nanosecond pulses, the size of pores is more likely in nanometres than in micrometres.
6. Figure 5d: The scale of y axis is confusing.
7. Figure 6c: Both graphs can be combined in one.
8. Results (line 408): Can you justify this part of the sentence: "...the presence of extracellular Ca^{2+} rather enhanced the loss of spheroid stiffness. " Is the difference in stiffness after EP in each media (for each cell type) significant?
9. Results (line 403-411): This paragraph belongs more to the Discussion part.
10. Materials and Methods: Please specify the referred figures from the Supplement (a, b, c etc.) since it is not always obvious which one from the composite figures to look at (e.g., line 635).
11. Materials and Methods (line 721): Describe how the form factor was determined.

Version 1:

Reviewer comments:

Reviewer #1

(Remarks to the Author)

The authors responded sincerely to the reviewers' comments and appropriately revised the text and figures in the manuscript. The revised manuscript is a significant improvement over the initial submission. I agree to its acceptance and publication in the journal.

Reviewer #2

(Remarks to the Author)

COMMENTS FOR THE AUTHORS R1

I found the enclosed revised manuscript COMMSBIO-25-6484-T entitled "Exposure to Nanosecond Pulsed Electric Fields Reveals Increased Cell Membrane Vulnerability and Impaired Cell Membrane Repair in Urothelial Cancer Cells." by Kielbik et al. suitable for publication in Communications Biology after minor corrections are made.

The authors have made a significant improvement of the manuscript. The majority of my comments and suggestions were taken into account. However, there are still a minor issue to be resolved:

1. Fig. 1a sub-ROI is still from left to right (the image is rotated by 90 degrees as seen on simulations in Figures 1b and S1b) and the simulations in Fig. 1b and S1b are upside down. All the figures about this setup have to be oriented uniformly to avoid confusion.

Reviewers' comments:

Reviewer #1 (Remarks to the Author):

This study describes an attempt to develop a targeted cancer therapy method using a nanosecond pulsed electric field (nsPEF) to induce membrane permeability. Using a fluorescence dye uptake assay with two-dimensional cultured urothelial cells, the authors demonstrated that nsPEF-induced membrane permeability is more selective toward cancer cells than toward normal cells. Furthermore, patient-derived organoids were used to show that nsPEFs induce significant swelling and reduced stiffness in cancer cells. The presented data suggest the therapeutic potential of nsPEFs for treating malignant urothelial cancer.

This study is technically sound, and the presented data are compelling. However, several concerns remain, such as the use of different cell lines in two-dimensional (2D) culture and organoid formation, as well as the lack of direct comparisons between experimental conditions. Furthermore, the relationship between the results obtained from organoids and the claims made in this paper is unclear.

Response:

Thank you for this important and constructive comment. To address the reviewer's concern and link the results obtained at the single-cell level with the observations in spheroids, we performed an additional analysis of the size of individual cells within the spheroids, using the same approach as applied for cell size evaluation in tissue samples. The corresponding H&E-stained images of the spheroids have been added to the Supplementary Materials, and the quantitative analysis of cell size is now presented in Figure 5e.

The analysis of single cells within the spheroids confirmed our previous observations:

- a) Healthy urothelial cells exhibited the smallest cell size among all spheroid types.
- b) To assess the relationship between cell size and electroporation efficiency, we performed a correlation analysis between the mean single-cell diameter within each spheroid and the corresponding fluorescence intensity. Linear regression analysis revealed a statistically significant positive correlation between mean cell size and YO-PRO-1 uptake ($r = 0.36$, $r^2 = 0.13$, $p = 0.0148$) (Figure 5d). Despite substantial variability in fluorescence among spheroids of the same type, spheroids composed of larger cells consistently showed higher permeabilization levels. These findings corroborate our single-cell data, demonstrating that cell size contributes to electroporation efficiency, although additional factors beyond cell size clearly influence spheroid permeability. A significant correlation with cell size was observed only when electroporation was conducted in a solution containing 2 mM Ca^{2+} , corresponding to physiological conditions. In contrast, no significant correlation was detected at either 0 mM or 5 mM Ca^{2+} , indicating that both Ca^{2+} depletion and elevated Ca^{2+} concentrations attenuate the apparent dependence of electroporation efficiency on cell size.

We write in the manuscript:

YP uptake findings align with observations from cell monolayer permeabilization studies, highlighting that nsPEFs induce significantly greater membrane disruption in urothelial cancer cells than in non-malignant counterparts. Linear regression analysis revealed a statistically significant positive correlation between mean cell size and YP uptake after exposure in 2 mM Ca^{2+} solution ($r = 0.36$, $r^2 = 0.13$, $p = 0.0148$) (Figure 5d). Despite considerable variability in fluorescence intensity among spheroids of the same type, spheroids composed of larger cells consistently exhibited higher levels of permeabilization. No significant correlation

was observed at 0 or 5 mM Ca^{2+} , suggesting that extremes of extracellular calcium attenuate the size-dependent effect on membrane permeabilization, through cell type-specific interactions between Ca^{2+} and membrane repair.

The differences in YP uptake observed between various spheroids and PDOs could not be attributed to variations in whole spheroid size. Linear regression analysis revealed no significant correlation between spheroid diameter and YP fluorescence intensity measured 10 minutes after exposure at any of the tested Ca^{2+} concentrations (supplementary Fig. S8e). These results show that, in three-dimensional spheroids, membrane permeabilization is governed primarily by the size of individual cells rather than by the overall spheroid dimensions.

Performing the single-cell size analysis within spheroids strengthens the link between our observations from 2D cultures and those obtained from spheroids derived from cell lines and patient samples. To reflect this, we now present the single-cell size analysis in the main manuscript figure, while the correlation between whole-spheroid size and permeabilization as well as influence of Ca^{2+} on spheroid permeabilization has been moved to the Supplementary Data.

2. Figure 5d shows that YP uptake into RT4 cells increases with 5 mM Ca^{2+} compared to 2 mM. This behavior differs from that of other cells, and possible reasons should be discussed.

Thank you for this valuable comment. Indeed, our data show that the permeabilization of RT4 spheroids was slightly higher under 5 mM Ca^{2+} compared to 2 mM Ca^{2+} conditions. The most pronounced differences were generally observed between samples electroporated with and without extracellular Ca^{2+} , while the differences between 2 mM and 5 mM Ca^{2+} were less distinct. Nevertheless, as the reviewer correctly pointed out, RT4 spheroids exhibited stronger YP uptake at 5 mM Ca^{2+} than at 2 mM Ca^{2+} .

This observation may be related to the dual effect of Ca^{2+} influx, which can both promote membrane resealing and influence the cell swelling process. In the RT4 cell line, electroporation in 5 mM Ca^{2+} resulted in significantly reduced cell swelling, as evidenced by a smaller increase in the 2D projected area of the spheroid following electroporation. Reduced swelling—compared to exposure in 0 mM or 2 mM Ca^{2+} buffers—may lead to fluorescence signals accumulating within a smaller area, thereby increasing the mean fluorescence intensity within the 2D projection.

We expect the influence of cell swelling on fluorescence intensity to be minor. The overall effect of extracellular calcium varied between spheroid and PDO types, and fluorescence intensity generally decreased with increasing Ca^{2+} concentration in all tested samples. Only RT4 spheroids showed a slight, unexpected increase in fluorescence at 5 mM Ca^{2+} compared to 2 mM Ca^{2+} , which might be partially explained by the reduced swelling effect described above.

We described this observation results:

All PDOs and spheroids derived from RT4 and SV-HUC-1 cells exhibited a Ca^{2+} -dependent decrease in YP fluorescence (supplementary Fig. S8d). On average, fluorescence measured 10 min after exposure decreased by 1.6-fold in the presence of 2 mM Ca^{2+} and by 1.8-fold with 5 mM Ca^{2+} . Notably, increasing Ca^{2+} from 2 mM to 5 mM reduced fluorescence in all spheroids except RT4, which showed a 1.2-fold increase.

And added the paragraph to the discussion:

While cell swelling may have influenced the average fluorescence intensity within spheroids, this effect is expected to be minor. The effect of extracellular Ca^{2+} on cell swelling varied among spheroid and PDO types, whereas YP fluorescence intensity generally decreased as Ca^{2+} concentration increased. Only in RT4 spheroids did an increase in Ca^{2+} from 2 mM to 5 mM slightly enhance YP fluorescence intensity. In these spheroids, exposure to nsPEFs at 5 mM Ca^{2+} also resulted in a smaller increase in the 2D projected cell area compared with 2 mM Ca^{2+} . Thus, reduced swelling in RT4 spheroids may have confined fluorescence signals to a smaller area, leading to an apparent increase in mean fluorescence intensity within the 2D projection.

3. Figure 6a shows that urothelial cancer-derived organoids swell when treated with nsPEFs. Figures 6b–d also demonstrate that nsPEFs decrease organoid stiffness. These results reflect the degree of membrane permeabilization caused by nsPEFs and align with the YP uptake results presented in Figure. However, this paper does not discuss the therapeutic significance of cellular swelling or stiffness reduction. Therefore, the significance of the data presented in Figure 6 is unclear.

Thank you for pointing this out. As the reviewer correctly noted, cell swelling and loss of stiffness are subsequent events following membrane permeabilization. YO-PRO-1 uptake is a well-established marker of plasma membrane permeabilization. However, to further strengthen the evidence for selective permeabilization, we also assessed secondary effects associated with membrane disruption.

We added a paragraph to the discussion:

Disassembly of actin structures and loss of cellular stiffness are known secondary effects of cell swelling induced by membrane permeabilization following nsPEF exposure. In this study, we demonstrated that the extent of spheroid stiffness loss corresponds to the differential severity of membrane disruption, with spheroids derived from normal SV-HUC-1 cells being less affected than those from malignant RT4 cells. Accordingly, urothelial cancer cells exhibited not only higher YP uptake but also more pronounced immediate secondary effects of membrane permeabilization.

Among the observed parameters, YO-PRO-1 uptake represents the most direct measure of membrane permeabilization. Osmotic swelling and the loss of cellular stiffness are additionally influenced by cytoskeletal integrity. The secondary effects of reduced stiffness and uncontrolled swelling likely contribute to cell death, which may occur as a delayed consequence of membrane permeabilization. Nevertheless, cell death following electroporation is far more complex and depends on multiple additional factors, such as the cell's Ca^{2+} handling capacity and resistance to ATP depletion. Addressing these complexities lies beyond the scope of the present study, which focuses on membrane permeabilization and its immediate secondary effects. This aspect is further discussed in the Discussion section of the manuscript.

Although electroporation is a transient process, it may ultimately lead to cell death due to Ca^{2+} overload, mitochondrial dysfunction, ROS production, and ATP depletion^{49–52}. Therefore, while extracellular Ca^{2+} enhance membrane resealing, it may also promote delayed cell death⁵³. The overall cytotoxic effect may thus depend on cell's capacity to restore intracellular Ca^{2+} levels or osmotic balance, and not exactly mirror the initial extent of membrane damage alone⁵⁴.

4. Lines 407–408 state that extracellular Ca^{2+} enhances the reduction in spheroid stiffness. However, Figure 6c shows that there was no significant difference between 0 mM and 2 mM Ca^{2+} in normal or cancer cells. Please explain the validity of this interpretation.

Thank you for this comment. We agree with the reviewer that, based on our data, no definitive conclusion can be drawn regarding the influence of extracellular Ca^{2+} on spheroid stiffness.

The earlier statement suggesting a Ca^{2+} -enhanced reduction in spheroid stiffness was based on results showing that, 10 minutes after exposure in 0 mM Ca^{2+} , the stiffness of SV-HUC-1 spheroids had decreased by 29%, whereas RT4 spheroids showed a 59% reduction. In 2 mM Ca^{2+} , SV-HUC-1 spheroids exhibited a 52% reduction in stiffness, while RT4 spheroids showed a 64% decrease within 10 minutes post-exposure.

However, statistical comparison of the absolute stiffness values following electroporation in 0 mM and 2 mM Ca^{2+} did not reveal a significant difference. Therefore, we have removed the statement claiming that Ca^{2+} enhances the reduction in spheroid stiffness from the manuscript.

5. The graph in Figure 5c does not appear to correspond to the YP uptake fluorescence images in Figure 5b. For instance, PDO270 seems to have the second-highest YP uptake after 319, but the graph always follows the trajectory closest to SV-HUC-1 (normal cell). Additionally, it seems that the colors in the graph legend are reversed for PDO154 and PDO270. Please confirm.

Thank you for this important comment. We have replaced the images of the electroporated PDOs to better represent the trajectory of YO-PRO-1 uptake in the spheroids. We agree with the reviewer that the presented graphs and images should accurately reflect the experimental data. The legend of the graphs was correct; however, the previously shown images were not fully representative of the overall YO-PRO-1 uptake dynamics.

Minor points:

1. In Figure 2b, the colors in the graph legend are reversed for 0 mM Ca^{2+} and 10 kV/cm, as well as for 2 mM Ca^{2+} and 10 kV/cm. This needs to be corrected.

Response: Thank you for this notion. The colors in the graph legend were indeed reversed and this mistake was corrected in the reviewed version.

2. In line 289, "(Figure 5b)" should be "(Figure 5c)."

Response: Thank you for this notion we have changed the reference to the figure

3. In Figure 6d, "(sek)" should be "(sec)."

Response: Thank you for noticing this error. we have change x axis description to time after PEF [s] to use uniformly abbreviation s for seconds.

4. Line 94: Since the term "AUC" appears for the first time in the main text, the full spelling should be provided.

Response: Thank you for noticing this error. We have spelled clarified the description.

5. Lines 288–304, 311–316, and 340–363 offer thorough explanations of the numerical values

depicted in the graphs. However, these explanations are overly verbose. To improve readability, please revise the text to provide brief summaries of the key points.

Response: Thank you for this comment. We modified the paragraphs to avoid describing the numerical values depicted in the graphs but to describe the key data necessary to understand the main observation from the experiments:

We have modified lines **288–304**:

At 10 minutes post-exposure in 0 mM Ca²⁺, the final YP fluorescence level in all tested PDOs was 2-3 times higher than that measured in spheroids derived from normal urothelial SV-HUC-1 cells ($p < 0.0001$ for all comparisons with SV-HUC-1 spheroids) (Figure 5c. supplementary Fig. S8b,c). A similar trend was observed in the presence of 2 mM Ca²⁺ with final fluorescence 1.5 to 3.5 times higher in PDO compared to spheroids from normal urothelial cells ($p < 0.05$). Under 5 mM Ca²⁺ condition, PDO 270 showed a modest 1.2-fold higher YP uptake, which was not significantly different from normal cell spheroids ($p > 0.05$), whereas other PDOs still exhibited 1.8 to 3.9 to times higher YP fluorescence than SV-HUC-1 cells ($p < 0.0001$).

Lines 311–316:

All PDOs and spheroids derived from RT4 and SV-HUC-1 cells exhibited a Ca²⁺-dependent decrease in YP fluorescence (supplementary Fig. S8d). On average, fluorescence measured 10 min after exposure decreased by 1.6-fold in the presence of 2 mM Ca²⁺ and by 1.8-fold with 5 mM Ca²⁺. Notably, increasing Ca²⁺ from 2 mM to 5 mM reduced fluorescence in all spheroids except RT4, which showed a 1.2-fold increase.

Lines 340– 363:

Specifically, the 2D projected area of PDOs in 0 mM Ca²⁺ increased from 14 to 21 % whereas in RT4 spheroids the area expanded by 42%. In contrast, spheroids from normal urothelial SV-HUC-1 cells showed only 12% increase in 2D projected area— significantly lower than all PDOs and RT4 spheroids ($p < 0.001$ for all comparisons with SV-HUC-1 spheroids). Also in the presence of 2 mM and 5mM Ca²⁺, all urothelial cancer spheroid types exhibited a significantly greater size increase compared to non-malignant SV-HUC-1 spheroids ($p < 0.0001$ for all comparisons).

Reviewer #2 (Remarks to the Author):

Electroporation have been successfully used in many applications in medicine, such as treating several types of cancer, drug delivery, gene therapy and cardiac arrhythmia. Therefore, exploring new possibilities of medical treatments is of significant importance. In this research, the authors study the mechanisms of electroporation effects on normal and cancer urothelial cells for developing treatment of urinary bladder cancer. The study is very well designed and executed, with great care for details. The statistical analysis is appropriate. The results are efficiently presented and discussed. There are only minor issues to be resolved, mainly in the presentation of the results.

Here are my comments and suggestions:

1. Results (line 94): Define the abbreviation AUC on the first appearance.

Thank you for your comment. We have now defined the abbreviation AUC at its first occurrence in the manuscript.

2. Supplement, Figures S2 and S3: From Figures S2b and S3 (the first in a and last in b) it can be concluded that images (the modelled one and the microscopic one) are in the opposite direction. It would be better to match the model with the experimental image. Moreover, if the sub-ROI in S3b match the regions of E field in S2b, different E should also be noted in S3b (last image) for greater clarity. The same goes for Figures 1a, b.

Thank you for this comment. We have adjusted the orientation of the simulated electric field figure to align with the microscopic images of YO-PRO-1 uptake. Furthermore, the calculated electric field values have been added to the microscopic images in the supplementary data (Supplementary Figures S1 and S11) as well as in Figure 1a to improve clarity.

3. Figure 1b: It is not clear at which axis the graph of E was determined. Why is half of it black and half of it green. What does it mean "across the region of interest"?

Thank you for pointing this out. To improve clarity, we have marked the dashed line between the electrodes in the same color used in the corresponding graph showing the data on Electric field distribution. Furthermore, we revised the wording "across the region of interest" to the more precise description "along the line perpendicular to the axis connecting the centers of both electrodes."

4. Figure 2a: What is the cell extension on the second image of the schematic?

The extensions illustrate water molecules entering the nanopores formed in the cell membrane. To enhance clarity, we have revised the corresponding description in the figure legend:

(2) Following electroporation, transient nanopores form in the plasma membrane, enabling the entry of YP and water from the extracellular environment into the cell. The kinetics of dye uptake can be fitted with a single-phase exponential curve, reflecting the dynamics of membrane permeability

5. Figure caption 2 (line 954): Considering the nanosecond pulses, the size of pores is more likely in nanometres than in micrometres.

Thank you for raising this point. We agree with the reviewer regarding the size of the membrane defects induced by PEF exposure and have replaced the term micropores with nanopores in the description.

6. Figure 5d: The scale of y axis is confusing.

Thank you for this comment. We have modified the graph to better illustrate the Ca^{2+} -dependent changes in the fluorescence AUC, including individual data points obtained from the spheroid measurements. The axis has also been adjusted to start at zero to avoid confusion. The graph has been moved to the supplementary data (Figure S9d).

7. Figure 6c: Both graphs can be combined in one.

Thank you for this observation. We decided to present the two graphs separately, as the spheroid stiffness was analyzed independently for RT4 and SV-HUC-1 cells, both with and without Ca^{2+} . The comparison of stiffness before and at different time points after exposure is shown in Figure 6d.

8. Results (line 408): Can you justify this part of the sentence: "...the presence of extracellular Ca²⁺ rather enhanced the loss of spheroid stiffness. " Is the difference in stiffness after EP in each media (for each cell type) significant?

Thank you for this comment. We agree with the reviewer that, based on our data, no definitive conclusion can be drawn regarding the influence of extracellular Ca²⁺ on spheroid stiffness.

The earlier statement suggesting a Ca²⁺-enhanced reduction in spheroid stiffness was based on results showing that, 10 minutes after exposure in 0 mM Ca²⁺, the stiffness of SV-HUC-1 spheroids had decreased by 29%, whereas RT4 spheroids showed a 59% reduction. In 2 mM Ca²⁺, SV-HUC-1 spheroids exhibited a 52% reduction in stiffness, while RT4 spheroids showed a 64% decrease within 10 minutes post-exposure.

However, statistical comparison of the absolute stiffness values following electroporation in 0 mM and 2 mM Ca²⁺ did not reveal a significant difference. Therefore, we have removed the statement claiming that Ca²⁺ enhances the reduction in spheroid stiffness from the manuscript.

9. Results (line 403-411): This paragraph belongs more to the Discussion part.

Thank you for this comment. We have moved this paragraph to the discussion part:

Disassembly of actin structures and loss of cellular stiffness are known secondary effects of cell swelling induced by membrane permeabilization following nsPEF exposure⁴³. In this study, we demonstrated that the extent of spheroid stiffness loss corresponds to the differential severity of membrane disruption, with spheroids derived from normal SV-HUC-1 cells being less affected than those from malignant RT4 cells. Accordingly, urothelial cancer cells exhibited not only higher YP uptake but also more pronounced immediate secondary effects of membrane permeabilization.

10. Materials and Methods: Please specify the referred figures from the Supplement (a, b, c etc.) since it is not always obvious which one from the composite figures to look at (e.g., line 635).

Thank you for this comment. We have revised the references to the supplementary figures as follows: (1) the figures have been reordered to match their sequence of appearance in the manuscript, and (2) each reference now specifies the corresponding panel.

11. Materials and Methods (line 721): Describe how the form factor was determined.

Thank you for this comment. We have added a detailed description of how the form factor was determined to the manuscript:

After cell segmentation, CellProfiler calculated the form factor for each identified object based on its shape parameters (form factor = $4\pi \times (\text{area} \times \text{perimeter}^{-2})$). This value ranges from 0 to 1, where 1 represents a perfect circle and lower values indicate increasingly elongated or irregular shapes.